# ADOPT: Modified Adam Can Converge with the Optimal Rate with Any Hyperparameters

## Abstract

Adaptive gradient methods based on exponential moving averages, such as `Adam` and `RMSprop`, are widely used for deep learning. However, it is known that they do not converge unless choosing hyperparameters in a problem-dependent manner. There have been many attempts to fix their convergence (e.g., `AMSGrad`), but they require an impractical assumption that the stochastic gradient is uniformly bounded. In this paper, we propose a new adaptive gradient method named `ADOPT`, which achieves the optimal convergence rate of $\mathcal{O}(1/\sqrt{T})$ with any hyperparameter choice without the bounded stochastic gradient assumption. `ADOPT` addresses the non-convergence issue of `Adam` by removing the current gradient from the second moment estimate and changing the order of the momentum calculation and the scaling operation by the second moment estimate. We also conduct intensive numerical experiments, and verify that our `ADOPT` achieves competitive or even better results compared to `Adam` and its variants across a wide range of tasks, including image classification, generative modeling, natural language processing, and deep reinforcement learning.

## 1 Introduction

Stochastic optimization algorithms, such as stochastic gradient descent (SGD), play a central role in deep learning. In particular, adaptive gradient methods based on exponential moving averages are widely used in practice. Despite the empirical success, it is known that some of the most popular algorithms, including Adam (Kingma & Ba, 2014) and RMSprop (Hinton et al., 2012), do not converge in theory. For example, Reddi et al. (2018) show that Adam and RMSprop fail to converge to a correct solution in a simple example where the objective function at time $t$ is given as:

$$f_t(\theta) = \begin{cases} C\theta, & \text{for } t \bmod 3 = 1 \\ -\theta, & \text{otherwise,} \end{cases} \tag{1}$$

where $C > 2$ and $\theta \in [-1, 1]$. In this online optimization setting, Adam and RMSprop with specific hyperparameters converge to a wrong solution (i.e., $\theta = 1$) instead of the true solution (i.e., $\theta = -1$). There have been several attempts to fix the non-convergent behavior of Adam (Reddi et al., 2018; Zou et al., 2019). For example, AMSGrad (Reddi et al., 2018) ensures the convergence for online convex optimization by making slight modifications to the Adam algorithm. Subsequent studies (Chen et al., 2019; Zhou et al., 2018) show that AMSGrad also converges to a stationary point for smooth nonconvex stochastic optimization problems. However, the convergence proofs rely on the assumption that the stochastic gradient is uniformly bounded. This assumption is stronger than the one used for the analysis of vanilla SGD (Ghadimi & Lan, 2013; Bertsekas & Tsitsiklis, 2000; Khaled & Richtárik, 2023), and is often violated in practice. For example, when Gaussian noise is used in the gradient estimation (e.g., the reparameterization trick in variational autoencoders (Kingma & Welling, 2014)), the stochastic gradient is no longer bounded.

Concurrently, Zhou et al. (2019) analyze the cause of non-convergence in Adam and RMSprop in the problem described in Eq. (1) from the perspective of the correlation between the current gradient and the second moment estimate based on the exponential moving average. Specifically, they show that the issue can be resolved by excluding the gradient of the most recent $n$ steps from the calculation of the second moment estimate, where $n$ is a hyperparameter that is equal to or larger than 1. They extend the analysis to the case where momentum is incorporated, as in Adam, and

propose AdaShift, which calculates momentum using only the gradient of the most recent $n$ steps to ensure that momentum is uncorrelated with the second moment estimate. However, their theoretical analysis is limited to a single online convex problem described in Eq. (1), and the convergence of AdaShift for general nonconvex problems is unclear. Moreover, this approach involves a trade-off in the choice of $n$: when $n$ is small, momentum has limited information about past gradients, and when $n$ is large, the second moment estimate has limited information about recent gradients.

More recently, some works have demonstrated that Adam can converge by choosing the hyperparameters in a problem-dependent manner (Shi et al., 2020; Zhang et al., 2022; Wang et al., 2022; Li et al., 2023). However, tuning the hyperparameters for each specific problem is troublesome; hence developing algorithms with the problem-independent convergence guarantee is still important to safely apply adaptive gradient methods to general machine learning problems.

In this paper, we propose an alternative approach to addressing the non-convergence issue of Adam without encountering trade-offs in hyperparameters or relying on strong assumptions such as the bounded stochastic gradient assumption. To derive our algorithm, we first examine the case without momentum, analyzing the convergence bound of RMSprop for general smooth nonconvex optimization problems. Through this analysis, we uncover the fundamental cause of divergence, which stems from the correlation between the second moment estimate and the current gradient. This finding aligns with the results demonstrated by Zhou et al. (2019) for online convex optimization. To resolve the divergence problem, we introduce slight modifications to the RMSprop algorithm that eliminate the correlation. Subsequently, we extend our findings to the case where momentum is incorporated, as in Adam, and discover that the Adam-style momentum also contributes to non-convergence. Although AdaShift addresses this issue by removing past gradients from momentum, it introduces a trade-off as previously described. However, we propose a modification that overcomes this trade-off by changing the order of the momentum calculation and the scaling operation using the second moment estimate. With this small adjustment, we successfully eliminate the non-convergence problem of Adam without relying on a specific hyperparameter choice and the bounded stochastic gradient assumption. We provide theoretical evidence demonstrating that our derived algorithm, named ADaptive gradient method with the OPTimal convergence rate (ADOPT), can achieve convergence with the optimal rate of $\mathcal{O}(1/\sqrt{T})$ for smooth nonconvex optimization.

In our experiments, we begin by assessing the performance of ADOPT in a toy example where Adam typically fails to converge. This toy example is an extension of the one presented in Eq. (1) by Reddi et al. (2018), but we consider a scenario where the bounded stochastic gradient assumption does not hold. Our results demonstrate that ADOPT rapidly converges to the solution, while Adam fails to converge, and AMSGrad exhibits slow convergence due to the violation of the assumption. Next, we conduct an experiment using a simple multi-layer perceptron on the MNIST classification task to evaluate the performance of ADOPT in nonconvex optimization. Our findings indicate that ADOPT outperforms existing adaptive gradient methods, including Adam, AMSGrad, and AdaShift. Finally, we evaluate the performance of ADOPT in various practical applications, such as ImageNet classification using modern neural networks (SwinTransformer), training of deep generative models (NVAE), fine-tuning of language models (LLaMA), and deep reinforcement learning for continuous control. Our empirical results demonstrate that ADOPT achieves competitive or even superior results over existing algorithms (e.g., Adam) in these practical applications.

## 2 PRELIMINARY

### 2.1 PROBLEM DEFINITION

We consider the minimization of the objective function $f : \mathbb{R}^D \to \mathbb{R}$ with respect to the parameter $\boldsymbol{\theta} \in \mathbb{R}^D$. In this context, we focus on first-order stochastic optimization methods, where only the stochastic gradient $\boldsymbol{g}$ is accessible. As the objective $f$ can be nonconvex, the goal is to find a stationary point where $\nabla f(\boldsymbol{\theta}) = 0$ (Blair, 1985; Vavasis, 1995). In order to analyze the convergence behavior of stochastic optimization algorithms, we adopt the following assumptions commonly employed in the literature[1] (Ghadimi & Lan, 2013; Zou et al., 2019; Défossez et al., 2022):

**Assumption 1.** *The objective function $f(\boldsymbol{\theta})$ is lower-bounded, i.e., $f(\boldsymbol{\theta}) \geq f_{\inf} > -\infty$ for all $\boldsymbol{\theta}$.*

---

[1]Note that Assumption 4 is often relaxed to an assumption that the variance (instead of the second moment) of the stochastic gradient is uniformly bounded, but we adopt Assumption 4 for the simplicity of our proofs.

**Assumption 2.** *The stochastic gradient $g_t$ is an unbiased estimator of the objective $f(\theta_{t-1})$, i.e., $\mathbb{E}[g_t] = \nabla f(\theta_{t-1})$ for all $t \geq 1$.*

**Assumption 3.** *The objective function is $L$-smooth on $\Theta$, i.e., there exists a constant $L > 0$ such that $\|\nabla f(x) - \nabla f(y)\| \leq L\|x - y\|$ for all $x, y \in \Theta$.*

**Assumption 4.** *The stochastic gradient has a finite second moment, i.e., there exists a constant $G > 0$ such that $\mathbb{E}[\|g_t\|^2] \leq G^2$.*

In the literature on convergence analysis, it is common to analyze the convergence rate of $\min_t\{\mathbb{E}[\|\nabla f(\theta_t))\|^2]\}$, where $\theta_t$ represents the parameter value after $t$ parameter updates.

For the analysis of adaptive gradient methods (e.g., Adam and AMSGrad), many of previous works (Chen et al., 2019; Zhou et al., 2018; Défossez et al., 2022) make an additional assumption that the stochastic gradient $g_t$ is uniformly bounded:

**Assumption 5.** *The stochastic gradient is uniformly upper-bounded, i.e., there exists a constant $G > 0$ such that $\|g_t\| \leq G$.*

Note that when Assumption 5 holds, Assumption 4 is automatically satisfied. Therefore, Assumption 5 is a stronger assumption compared to Assumption 4. When we omit Assumption 5, it becomes challenging to analyze $\min_t\{\mathbb{E}[\|\nabla f(\theta_t))\|^2]\}$ for adaptive gradient methods. As a result, the analysis often considers $\min_t\{\mathbb{E}[\|\nabla f(\theta_t))\|^{4/3}]^{3/2}\}$ instead. In this paper, we focus on analyzing $\min_t\{\mathbb{E}[\|\nabla f(\theta_t))\|^{4/3}]^{3/2}\}$, because one of our motivations is to address the omission of Assumption 5.

## 2.2 REVIEW OF STOCHASTIC OPTIMIZATION ALGORITHMS FOR NONCONVEX OBJECTIVES

The convergence of the vanilla SGD have been studied extensively in previous works. For smooth nonconvex functions, Ghadimi & Lan (2013) showed that SGD with a constant learning rate converges with an $\mathcal{O}(1/\sqrt{T})$ rate by setting $\alpha_t = \alpha = \Theta(1/\sqrt{T})$, where $\alpha_t$ is a learning rate at the $t$-th step, and $T$ is a total number of parameter updates. This convergence rate is known to be minimax optimal up to a constant (Drori & Shamir, 2020). For the diminishing learning rate scheme, the convergence bound of $\mathcal{O}(\log T/\sqrt{T})$ is well-known for $\alpha_t = \alpha/\sqrt{t}$ (Ghadimi & Lan, 2013). Recently, Wang et al. (2021) have proved that SGD with $\alpha_t = \alpha/\sqrt{t}$ can also achieve the optimal rate $\mathcal{O}(1/\sqrt{T})$ by additionally assuming that the objective $f$ is upper-bounded.

While the vanilla SGD is still one of the most popular choices for stochastic optimization, adaptive gradient methods are dominantly used especially for deep learning. In adaptive gradient methods, the parameter $\theta$ is updated additionally using the second moment estimate $v_t$ in the following form:

$$\theta_t = \theta_{t-1} - \alpha_t \frac{g_t}{\sqrt{v_t + \epsilon^2}}, \tag{2}$$

where $\epsilon$ is a small constant, the division between vectors is applied in an element-wise manner, and the addition between a vector $a$ and a scalar $b$ is defined as $(a + b)_i := a_i + b$. In AdaGrad (Duchi et al., 2011), $v_t$ is defined as $v_0 = 0$ and $v_t = v_{t-1} + g_t \odot g_t$. In RMSprop (Hinton et al., 2012), an exponential moving average is substituted for the simple summation, i.e., $v_t = \beta_2 v_{t-1} + (1 - \beta_2) g_t \odot g_t$, where $0 \leq \beta_2 < 1$. Adam (Kingma & Ba, 2014) uses momentum in addition to the second moment estimate to accelerate the convergence as follows:

$$\begin{cases} m_t = \beta_1 m_{t-1} + (1 - \beta_1) g_t, \\ \theta_t = \theta_{t-1} - \alpha_t \frac{m_t}{\sqrt{v_t + \epsilon^2}}, \end{cases} \tag{3}$$

where $m_0 = 0$. Here, we omit the bias correction technique used in the original paper for clarity. Unfortunately, RMSprop and Adam are not guaranteed to converge even in a simple convex optimization problem as demonstrated by Reddi et al. (2018), whereas AdaGrad with a constant learning rate is known to converge with an $\mathcal{O}(\log T/\sqrt{T})$ rate under Assupmtions 1-4 for smooth nonconvex cases (Li & Orabona, 2019; Ward et al., 2020; Zou et al., 2018; Chen et al., 2019; Défossez et al., 2022). Although the convergence of Adam can be assured by choosing the hyperparameters (i.e., $\beta_1$ and $\beta_2$) in a problem-dependent manner (Shi et al., 2020; Zhang et al., 2022; Wang et al., 2022; Li et al., 2023), it is difficult to know the proper hyperparameters for each problem before training. To fix the non-convergence of Adam without depending on a hyperparameter choice, some researchers

have proposed variants of Adam. Reddi et al. (2018) proposed AMSGrad, which substitute $\tilde{\boldsymbol{v}}_t$ for $\boldsymbol{v}$ in Eq. (3), where $\tilde{\boldsymbol{v}}_0 = \boldsymbol{0}$ and $\tilde{\boldsymbol{v}}_t = \max\{\tilde{\boldsymbol{v}}_{t-1}, \boldsymbol{v}_t\}$. The idea behind AMSGrad is that the scaling factor $\sqrt{\tilde{\boldsymbol{v}}_t + \epsilon^2}$ should be non-decreasing to ensure the convergence. After Reddi et al. (2018) originally proved the convergence of AMSGrad for online convex optimization, Chen et al. (2019) showed that AMSGrad with $\alpha_t = \alpha/\sqrt{t}$ converges with $\mathcal{O}(\log T/\sqrt{T})$ for nonconvex settings. Zhou et al. (2018) also analyzed the convergence of AMSGrad for nonconvex optimization, and derived the convergence rate of $\mathcal{O}(1/\sqrt{T})$ for a constant learning rate of $\alpha_t = \alpha = \Theta(1/\sqrt{T})$. However, their results depend on Assumption 5, which is often violated in practice. For example, variational autoencoders (Kingma & Welling, 2014) and diffusion models (Ho et al., 2020) are typical examples in which Assumption 5 does not hold because they utilize unbounded Gaussian noise in the gradient estimation. The cause of requirement for Assumption 5 is the max operation in the definition of $\tilde{\boldsymbol{v}}_t$. Since the max operation is convex, $\mathbb{E}[\tilde{\boldsymbol{v}}_t] \le \max_t\{\mathbb{E}[\boldsymbol{v}_t]\}$ does not hold; hence Assumption 5 is required to upper-bound $\mathbb{E}[\tilde{\boldsymbol{v}}_t]$ in their proofs. Zhou et al. (2019) also tried to fix the non-convergent behavior of Adam. Their proposed AdaShift uses $\boldsymbol{v}_{t-n}$ instead of $\boldsymbol{v}_t$ for the second moment estimate, and calculate the momentum using the latest $n$ gradients as follows:

$$
\begin{cases}
\boldsymbol{m}_t = \frac{\sum_{k=0}^{n-1} \beta_1^k \boldsymbol{g}_{t-k}}{\sum_{k=0}^{n-1} \beta_1^k}, \\
\boldsymbol{\theta}_t = \boldsymbol{\theta}_{t-1} - \alpha_t \frac{\boldsymbol{m}_t}{\sqrt{\boldsymbol{v}_{t-n} + \epsilon^2}}.
\end{cases}
\tag{4}
$$

In the original paper, some additional techniques (e.g., the block-wise adaptive learning rate) are used, but we omit them for clarity here. Though they give theoretical analysis for a single online convex example, any convergence bounds are not provided for nonconvex cases. More detailed discussion on related works is provided in Appendix A.

## 3    Cause of Non-convergence of Adam and How to Fix It

In this section, to derive an algorithm that can converge with any hyperparameter choice without the bounded stochastic gradient assumption, we analyze the cause of non-convergence of Adam, and discuss how it can be eliminated. To start from a simple case, we first analyze the case without momentum. Subsequently, we extend it to the case with momentum and provide a way to fix the convergence issue of Adam.

### 3.1    Case without Momentum

We first analyze the convergence of RMSprop, which corresponds to the no-momentum case of Adam when we omit the bias correction. For RMSprop, we derive the following convergence bound.

**Theorem 1.** *Under Assumptions 1, 2, 3, and 4, the following holds for the RMSprop with a constant learning rate $\alpha_t = \alpha$:*

$$
\min_{t=1,\ldots,T} \left\{ \mathbb{E}\left[\|\nabla f(\boldsymbol{\theta}_{t-1}))\|^{4/3}\right]^{3/2} \right\}
$$
$$
\le 2\sqrt{\left(1 - \beta_2^T\right) G^2 + \epsilon^2} \left( \frac{f(\boldsymbol{\theta}_0) - f_{\inf}}{\alpha T} + \frac{C}{T} \log\left(1 + \frac{\left(1 - \beta_2^T\right) G^2}{\epsilon^2}\right) - C \log\beta_2 \right),
\tag{5}
$$

*where $C = \frac{\alpha D L}{2(1 - \beta_2)} + \frac{2 D G}{\sqrt{1 - \beta_2}}$.*

*Sketch of proof.* By Assumption 3, the following holds:

$$
\mathbb{E}\left[f(\boldsymbol{\theta}_t)\right] \le \mathbb{E}\left[f(\boldsymbol{\theta}_{t-1})\right] - \alpha\mathbb{E}\left[\nabla f(\boldsymbol{\theta}_{t-1})^\top \left(\frac{\boldsymbol{g}_t}{\sqrt{\boldsymbol{v}_t + \epsilon^2}}\right)\right] + \frac{\alpha^2 L}{2}\mathbb{E}\left[\left\|\frac{\boldsymbol{g}_t}{\sqrt{\boldsymbol{v}_t + \epsilon^2}}\right\|^2\right]
\tag{6}
$$

Applying Lemmas 7 and 9 in the appendix to this, the following inequality is derived:

$$
\mathbb{E}\left[f(\boldsymbol{\theta}_t)\right] \le \mathbb{E}\left[f(\boldsymbol{\theta}_{t-1})\right] - \frac{\alpha}{2}\mathbb{E}\left[\nabla f(\boldsymbol{\theta}_{t-1})^\top \left(\frac{\boldsymbol{g}_t}{\sqrt{\tilde{\boldsymbol{v}}_t + \epsilon^2}}\right)\right] + \left(\frac{\alpha^2 L}{2} + 2\alpha G\sqrt{1 - \beta_2}\right)\mathbb{E}\left[\left\|\frac{\boldsymbol{g}_t}{\sqrt{\boldsymbol{v}_t + \epsilon^2}}\right\|^2\right]
\tag{7}
$$

$$
\le \mathbb{E}\left[f(\boldsymbol{\theta}_{t-1})\right] - \frac{\alpha}{2}\frac{\mathbb{E}\left[\|\nabla f(\boldsymbol{\theta}_{t-1})\|^{4/3}\right]^{3/2}}{\sqrt{\left(1 - \beta_2^T\right) G^2 + \epsilon^2}} + \left(\frac{\alpha^2 L}{2} + 2\alpha G\sqrt{1 - \beta_2}\right)\mathbb{E}\left[\left\|\frac{\boldsymbol{g}_t}{\sqrt{\boldsymbol{v}_t + \epsilon^2}}\right\|^2\right],
\tag{8}
$$

where $\tilde{\boldsymbol{v}}_t = \beta_2 \boldsymbol{v}_{t-1} + (1 - \beta_2)\mathbb{E}[\boldsymbol{g}_t \odot \boldsymbol{g}_t]$. Telescoping this for $t = 1, \ldots, T$ and rearranging the terms, we have

$$\sum_{t=1}^{T} \mathbb{E}\left[\|\nabla f\left(\boldsymbol{\theta}_{t-1}\right)\|^{4/3}\right]^{3/2}$$

$$\leq 2\sqrt{\left(1 - \beta_2^T\right) G^2 + \epsilon^2} \left(\frac{f\left(\boldsymbol{\theta}_0\right) - f_{\inf}}{\alpha} + C \log\left(1 + \frac{\left(1 - \beta_2^T\right) G^2}{\epsilon^2}\right) - CT \log \beta_2\right), \quad (9)$$

where the last inequality holds due to Assumption 1 and Lemma 8. Therefore, the bound in Eq. (5) is derived using the following fact:

$$\min_{t=1,\ldots,T} \left\{ \mathbb{E}\left[\|\nabla f(\boldsymbol{\theta}_{t-1}))\|^{4/3}\right]^{3/2} \right\} \leq \frac{\sum_{t=1}^{T} \mathbb{E}\left[\|\nabla f\left(\boldsymbol{\theta}_{t-1}\right)\|^{4/3}\right]^{3/2}}{T}. \quad (10)$$

$\square$

A detailed proof is provided in the appendix. When the learning rate $\alpha$ is chosen so that $\alpha = \Theta(1/\sqrt{T})$, the first and second terms on the right hand side of Eq. (5) converge with $\mathcal{O}(1/\sqrt{T})$ and $\mathcal{O}(1/T)$ rates, respectively. However, the last term includes a constant factor in terms of $T$, which represents the non-convergent behavior of RMSprop in the smooth nonconvex setting. More precisely, RMSprop is guaranteed to converge only to a bounded region around a stationary point, and the size of the bounded region depends on the hyperparameter $\beta_2$ and the problem-dependent factors $D$, $G$, and $L$. Therefore, we need to choose $\beta_2$ dependently on each problem to make the bounded region adequately small. Basically, the size of the bounded region can be made small by setting $\beta_2$ to a value close to 1, but how close to 1 it should be relies on the problem-dependent factors, which cannot be observed in advance. This result is consistent with recent results of convergence analyses of Adam and RMSprop (Shi et al., 2020; Zhang et al., 2022).

As can be seen from Eqs. (6) and (7), the constant term in Eq. (5) is derived from the second term of Eq. (6). Because $\boldsymbol{g}_t$ and $\boldsymbol{v}_t$ are not statistically independent, this term is first decomposed into the second and third terms of Eq. (7) by using Lemma 7. After the decomposition, $\boldsymbol{g}_t$ and $\tilde{\boldsymbol{v}}_t$ is now conditionally independent given $\boldsymbol{g}_0, \ldots, \boldsymbol{g}_{t-1}$, so Eq. (8) is derived using the following fact in Lemma 9:

$$\mathbb{E}\left[\frac{\boldsymbol{g}_t}{\sqrt{\tilde{\boldsymbol{v}}_t + \epsilon^2}}\right] = \mathbb{E}\left[\frac{\nabla f\left(\boldsymbol{\theta}_{t-1}\right)}{\sqrt{\tilde{\boldsymbol{v}}_t + \epsilon^2}}\right]. \quad (11)$$

In other words, if the second moment estimate is designed to be conditionally independent to $\boldsymbol{g}_t$, the constant term in the convergence bound will be removed, because the second term of Eq. (6) can be directly lower-bounded by a quantity propotional to $\mathbb{E}[\|\nabla f(\boldsymbol{\theta}_t))\|^{4/3}]^{3/2}$ as in Lemma 9. A simple way to achieve the conditional independence is to substitute $\boldsymbol{v}_{t-1}$ for $\boldsymbol{v}_t$ as a second moment estimate, because $\boldsymbol{v}_{t-1}$ does not have information about $\boldsymbol{g}_t$. This solution is similar to AdaShift, in which $\boldsymbol{v}_{t-n}$ is substituted for $\boldsymbol{v}_t$ as described in Eq. (4). In fact, the modified version of RMSprop is identical to AdaShift with $n = 1$ and $\beta_1 = 0$ except for the additional techniques (e.g., the block-wise adaptive learning rate).

## 3.2 CASE WITH MOMENTUM

As we have described, RMSprop can be modified to be convergent by removing the current gradient $\boldsymbol{g}_t$ from the second moment estimate $\boldsymbol{v}_t$. However, when we combine adaptive gradient methods with momentum like Adam, the convergence analysis becomes more complicated. Unfortunately, when Adam-style momentum in Eq. (3) is applied, the algorithm does not converge in general even when using $\boldsymbol{v}_{t-1}$ as a second moment estimate instead of $\boldsymbol{v}_t$. This is because the momentum $\boldsymbol{m}_t$ contains all history of the past gradients $\boldsymbol{g}_0, \ldots, \boldsymbol{g}_t$; hence the second moment estimate always correlates with $\boldsymbol{m}_t$. AdaShift prevents this problem by calculating the momentum $\boldsymbol{m}_t$ only using the latest $n$ gradients as described in Eq. (4). In that case, the momentum $\boldsymbol{m}_t$ and the second moment estimate $\boldsymbol{v}_{t-n}$ are conditionally independent, so the convergence can be retained. However, this approach has a trade-off in the choice of $n$. When $n$ is small, $\boldsymbol{m}_t$ has little information about the past gradients; when $n$ is large, $\boldsymbol{v}_{t-n}$ only has access to the gradient information in the distant past.

---

**Algorithm 1** ADOPT algorithm

---

**Require:** Learning rate $\{\alpha_t\}$, initial parameter $\boldsymbol{\theta}_0$
**Require:** Exponential decay rate $0 \leq \beta_1 < 1, 0 \leq \beta_2 \leq 1$, small constant $\epsilon \geq 0$
   $\boldsymbol{m}_0 \leftarrow \boldsymbol{0}, \boldsymbol{v}_0 \leftarrow \boldsymbol{1}$
   **for** $t = 1$ to $T$ **do**
      $\boldsymbol{m}_t \leftarrow \beta_1 \cdot \boldsymbol{m}_{t-1} + (1 - \beta_1) \frac{\boldsymbol{g}_t}{\sqrt{\boldsymbol{v}_{t-1}+\epsilon^2}}$
      $\boldsymbol{\theta}_t \leftarrow \boldsymbol{\theta}_{t-1} - \alpha_t \boldsymbol{m}_t$
      $\boldsymbol{v}_t \leftarrow \beta_2 \cdot \boldsymbol{v}_{t-1} + (1 - \beta_2) \boldsymbol{g}_t \odot \boldsymbol{g}_t$
   **end for**
   **return** $\{\boldsymbol{\theta}_t\}_{t=1}^T$

---

To remove this trade-off, instead of truncating the momentum to the latest $n$ steps, we propose to use momentum of the following form:

$$\begin{cases} \boldsymbol{m}_t = \beta_1 \boldsymbol{m}_{t-1} + (1 - \beta_1) \frac{\boldsymbol{g}_t}{\sqrt{\boldsymbol{v}_{t-1}+\epsilon^2}}, \\ \boldsymbol{\theta}_t = \boldsymbol{\theta}_{t-1} - \alpha_t \boldsymbol{m}_t, \end{cases} \tag{12}$$

The main difference to the Adam-style momentum in Eq. (3) is the order of momentum calculation and the scaling operation by $\sqrt{\boldsymbol{v}_{t-1} + \epsilon^2}$. In Eq. (3), the scaling operation is performed after the momentum calculation, whereas in Eq. (12), the scaling operation is first applied to the current gradient $\boldsymbol{g}_t$ in advance to the momentum calculation. In this case, the second moment estimate $\boldsymbol{v}_{t-1}$ is only used to scale the current gradient $\boldsymbol{g}_t$, so the convergence can be guaranteed. A more detailed convergence analysis is provided in Section 4.

## 4 ADOPT: Adaptive Gradient Method with the Optimal Convergence Rate

Based on the analysis in the previous section, we propose a new adaptive gradient method named ADOPT (ADaptive gradient method with the OPTimal convergence rate). The entire procedure is summarized in Algorithm 4. In ADOPT, to ensure the convergence, we use $\boldsymbol{v}_{t-1}$ as a second moment estimate instead of $\boldsymbol{v}_t$, and the scaling operation by $\sqrt{\boldsymbol{v}_{t-1} + \epsilon^2}$ is applied not to the momentum $\boldsymbol{m}_t$ but to the current gradient $\boldsymbol{g}_t$. To prevent the initial scaling factor $\sqrt{\boldsymbol{v}_0 + \epsilon^2}$ from being too small, we initialize $\boldsymbol{v}_0$ with 1 instead of 0. By this modification, ADOPT can converge with the optimal rate $\mathcal{O}(1/\sqrt{T})$ for the smooth nonconvex optimization as follows:

**Theorem 2.** *Under Assumptions 1, 2, 3, and 4, the following holds for the ADOPT algorithm with a constant learning rate $\alpha_t = \alpha$:*

$$\min_{t=1,\ldots,T} \left\{ \mathbb{E}\left[ \|\nabla f(\boldsymbol{\theta}_{t-1}))\|^{4/3} \right]^{3/2} \right\} \leq C_1(T) \left( \frac{f(\boldsymbol{\theta}_0) - f_{\inf}}{\alpha T} + \alpha C_2 \left( 1 - \frac{1}{T \log \beta_2} \log \left( \frac{\beta_2^T + \epsilon^2}{1 + \epsilon^2} \right) \right) \right), \tag{13}$$

*where* $C_1(T) = \sqrt{\max\left\{ G^2 + (1 - G^2)\beta_2^T, 1 \right\} + \epsilon^2}$, $C_2 = \frac{(1+\beta_1)G^2 L}{2(1-\beta_1)\epsilon^2}$.

The detailed proof and related lemmas are provided in the appendix. When we choose the learning rate so that $\alpha = \Theta(1/\sqrt{T})$, the right hind side of Eq. (13) converges with an $\mathcal{O}(1/\sqrt{T})$ rate. We also provide the convergence bound for the case of diminishing learning rate (i.e., $\alpha_t = \alpha/\sqrt{t}$) in the appendix, which is closer to practical situations. In that case, ADOPT also converges with the optimal rate of $\mathcal{O}(1/\sqrt{T})$.

## 5 Experiments

In the experiments, we first validate our ADOPT algorithm using a simple toy example in which Adam is known to fail to converge, and confirm our theoretical findings through numerical simulation. Secondly, we run an experiment of training a simple multi-layer perceptron (MLP) for the MNIST dataset to verify the effectiveness of our ADOPT for nonconvex optimization problems.

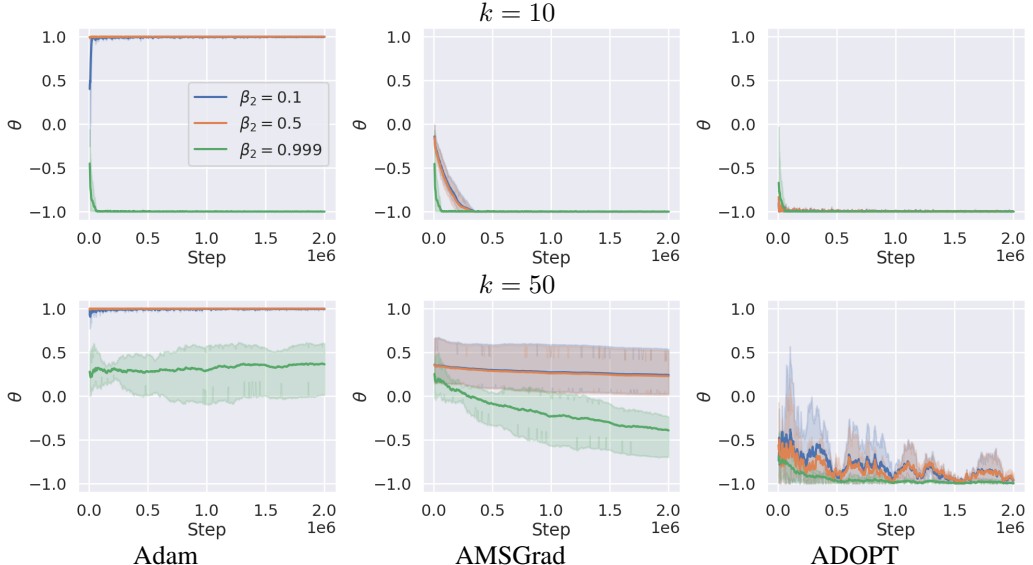

Figure 1: Performance comparison between Adam, AMSGrad and ADOPT in a simple univariate convex optimization problem. The plots show transitions of the parameter value, which should converge to the solution $\theta = -1$.

Finally, we evaluate our ADOPT in a wide range of practical applications, including image classification, natural language processing (NLP) tasks, generative modeling, and deep reinforcement learning. Detailed experimental settings are described in the appendix.

**Toy problem:** We consider a convex optimization problem with an objective $f(\theta) = \theta$ for $\theta \in [-1, 1]$. It is obvious that a solution for the problem is $\theta = -1$. Through the optimization, we only have access to the stochastic objective $f_t$ as follows:

$$f_t(\theta) = \begin{cases} k^2\theta, & \text{with probability } 1/k \\ -k\theta, & \text{with probability } 1 - 1/k \end{cases},$$ (14)

where $k \geq 1$. Because $\mathbb{E}[f_t(\theta)] = f(\theta)$ holds, the stochastic gradient $g_t = \nabla f_t(\theta)$ is an unbiased estimator of the true gradient $\nabla f$ regardless of the choice of $k$, satisfying Assumption 2. This problem is equivalent, except for scaling, to the stochastic optimization version of Eq. (1) provided by Reddi et al. (2018) as a case where Adam (and RMSprop) with specific hyperparameters fail to converge. The constant $k$ controls the magnitude of gradient noise. When $k = 1$, it corresponds to the noiseless case where $f_t = f$ with probability 1. As $k$ gets large, stochastic gradient becomes noisy, making $G$ in Assumptions 4 or 5 large. Therefore, the optimization will be more difficult when $k$ becomes larger. In the experiment, we set $k = 10$ or 50, and compare the robustness of Adam, AMSGrad, and ADOPT for various hyperparameter settings by changing $\beta_2$ from $0.1 \sim 0.999$. We set $\beta_1 = 0.9$ for all the algorithms, which is a common choice in practice. We set the learning rate to $\alpha_t = 0.01/\sqrt{1 + 0.01t}$. The parameter $\theta$ is initialized to 0 for all cases.

The result is shown in Figure 5. It can be seen that, when $k = 10$, Adam fails to converge except for $\beta_2 = 0.999$ while AMSGrad and ADOPT rapidly converge to the correct solution, i.e., $\theta = -1$. In a more extreme case where $k = 50$, Adam fails to converge even with $\beta_2$ very close to 1. This aligns with Theorem 1, since, when the gradient noise is large (i.e., $G$ is large), the bounded region of the convergence bound also gets large, leading to divergence of Adam. Moreover, when $k = 50$, it is observed that the convergence of AMSGrad also becomes much slower than ADOPT. In fact, this phenomenon is also consistent with theory. In this problem setting, the second moment $\mathbb{E}[g_t^2]$ is $\mathcal{O}(k^3)$, while the squared norm of the stochastic gradient $g_t^2$ is $\mathcal{O}(k^4)$. Since the convergence bound of AMSGrad depends on the uniform bound of the stochastic gradient in Assumption 5, instead of the second moment in Assumption 4, its convergence also deteriorates with the order of $g_t^2$. Compared to AMSGrad, ADOPT only depends on the second moment bound for its convergence, so it converges much faster than AMSGrad even in such an extreme setting, although the convergence speed depends on the choice of hyperparameters.

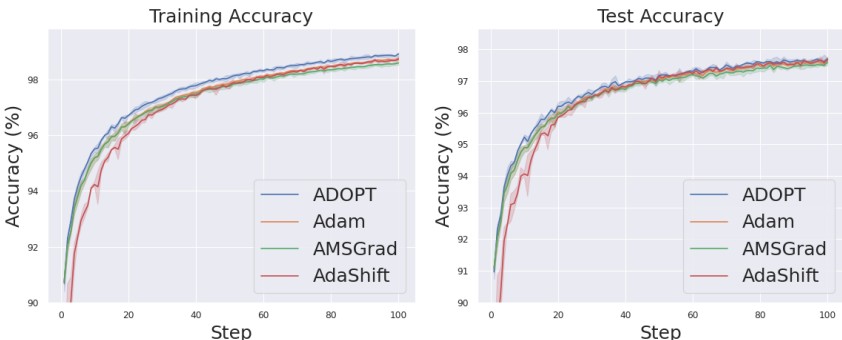

Figure 3: Accuracy for training data (left) and test data(right) in MNIST classification. The error bars show the 95% confidence intervals of three trials.

We also perform ablation study on how the two algorithmic changes from Adam to ADOPT affect the convergence. The differences between Adam and ADOPT are (1) decorrelation between the second moment estimate and the current gradient, and (2) change of order of momentum calculation and scaling operation by the second moment estimate. In this experiment, we remove each algorithmic change from ADOPT, and compare the result in the toy example. We set $k = 50$, and $(\beta_1, \beta_2) = (0.9, 0.999)$, since it is a common hyperparameter choice. The result is shown in Figure 2. It can be observed that ADOPT fails to converge with the exception of either algorithmic change. Therefore, applying both changes is essential to overcome the non-convergent issue of Adam, which also aligns with theory.

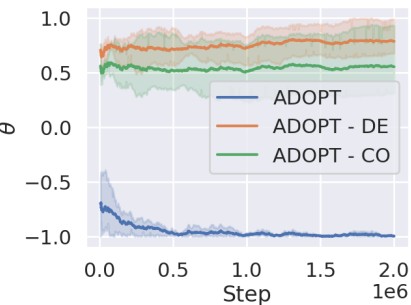

Figure 2: Ablation study of algorithmic changes between Adam and ADOPT. "DE" and CO denote "decorrelation" and "change of order", respectively.

These results correspond to the theoretical findings, showing the superiority of ADOPT to Adam and AMSGrad in terms of the convergence speed and its robustness to hyperparameter choices.

**MNIST classification:** To investigate the performance on nonconvex optimization, we compare ADOPT with Adam, AMSGrad and AdaShift, on the MNIST classification using an MLP with a single hidden layer. The number of hidden units is set to 784. We set the learning rate to $\alpha_t = \alpha/\sqrt{t}$, and $\alpha$ is tuned in the range of $\{1, 10^{-1}, 10^{-2}, 10^{-3}\}$. We apply weight decay of $1 \times 10^{-4}$ to prevent over-fitting, and run 10K iterations of parameter updates. Figure 3 shows the learning curves of training and test accuracy. We observe our ADOPT performs slightly better than the others in terms of the convergence speed and the final performance. Thanks to the way of the momentum calculation in Eq. (12), ADOPT works better than AdaShift especially in the early phase of training.

**ImageNet classification:** We perform ImageNet classification using SwinTransformer (Liu et al., 2021) to confirm that our ADOPT works well for modern vision Transformers. We follow the official training recipe of Swin Transformer-tiny provided by Torchvision[2], and fix the training settings except for the optimizer choice. We use AdamW (Loshchilov & Hutter, 2019) as a baseline because it is set as the default official optimizer. We also compare with AMSGrad as another way to fix the divergence issue of Adam. Since AdamW uses decoupled weight decay, we also apply it to the other optimizers for fair comparison. We report the top-1 accuracy at $\frac{T}{3}$, $\frac{2}{3}T$ and $T$ epochs in Tables 1, where $T$ is the total number of training epochs. We observe that ADOPT outperforms AdamW and AMSGrad throughout the training in terms of the test accuracy, demonstrating the effectiveness of ADOPT for this setting.

**Generative modeling:** We train NVAE (Vahdat & Kautz, 2020) for MNIST using our ADOPT. In the official implementation of NVAE, Adamax (Kingma & Ba, 2014), an infinite-norm variant of Adam, is used as an optimizer, so we use Adamax as a baseline method. We use the exactly the same

---

[2]https://github.com/pytorch/vision/tree/main/references/classification

Table 1: Top-1 accuracy (%) of SwinTransformer on ImageNet.

| Epoch | 200 | 300 |
|---|---|---|
| AdamW | $79.29 \pm 0.05$ | $81.26 \pm 0.04$ |
| AMSGrad | $78.91 \pm 0.03$ | $81.17 \pm 0.03$ |
| ADOPT | $\mathbf{79.62} \pm 0.03$ | $\mathbf{81.50} \pm 0.04$ |

Table 2: Negative log-likelihood of NVAEs for MNIST.

| Epoch | 200 | 300 |
|---|---|---|
| Adamax | $80.19 \pm 0.08$ | $79.41 \pm 0.07$ |
| ADOPT | $\mathbf{79.02} \pm 0.10$ | $\mathbf{78.88} \pm 0.09$ |

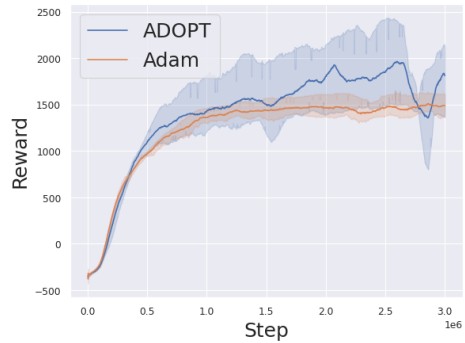

Figure 4: Performance comparison between Adam and ADOPT in reinforcement learning.

setting of the official implementation except that the learning rate for ADOPT is set to $2 \times 10^{-4}$ since the default value $0.01$ is too large for ADOPT. We report the negative log-likelihood for test data on Table 2. It is observed that the model trained with ADOPT shows the better likelihood.

**Finetuning of large language models:** We finetune the pretrained LLaMA-7B on 52K instruction-following data provided by Stanford Alpaca and compare the performance between the default optimizer (Adam,) and our ADOPT under the exactly same experimental setting. For evaluation, we use Multi-task Language Understanding (MMLU) Benchmark (Hendrycks et al., 2021), which is widely used to assess the performance of large language models. The MMLU score for LLaMA-7B without finetuning is $35.1$. After fine-tuned via instruction-following using the baseline implementation with Adam, the score improves to $41.2$. When we substitute ADOPT for Adam, the score even improves to $42.13$. The detailed score comparison for each task is summarized in Figure 5 in the appendix.

**Deep reinforcement learning:** Lastly, we train reinforcement learning (RL) agents using the proximal policy optimization (PPO) with ADOPT for the optimizer. As a benchmark, we use a continuous control tasks of HalfCheetah on MuJoCo simulator. For comparison to ADOPT, Adam is used as a baseline optimizer. We follow the hyperparameter settings recommended by Stable-Baselines3 (Raffin et al., 2021), and just change the choice of an optimizer. The result is shown in Figure 4. It can be observed that ADOPT shows competitive or even better performance than Adam.

## 6 LIMITATIONS

One of the limitations of our analysis is that it still relies on the assumption that the second moment of stochastic gradient is uniformly bounded (i.e., Assumption 4). Although this assumption is weaker than the bounded stochastic gradient assumption (i.e., Assumption 5), it would be more desirable to relax it to an assumption that the variance of the stochastic gradient is uniformly bounded, which is often adopted in the analysis of the vanilla SGD (Ghadimi & Lan, 2013). Extending our result to weaker assumptions is an important direction of future work.

## 7 CONCLUSION

In this paper, we demystified the fundamental cause of divergence of adaptive gradient methods based on the exponential moving average, such as Adam and RMSprop, in general smooth non-convex optimization problems, and demonstrate a way to fix the issue, proposing a new optimizer named ADOPT. Not only does ADOPT converge with the optimal rate without depending on a hyperparameter choice in theory, but ADOPT demonstrates competitive or even better results in the pracital applications, including ImageNet classification, generative modeling, finetuning of language models, and deep reinforcement learning. We expect that this work will serve as a bridge between theory and practice in the research of adaptive gradient methods. Since our ADOPT can be safely applied to many machine learning problems without careful tuning of hyperparameters, it can be expected to improve the training stability and the model performance in practice by substituting it for the existing adaptive gradien methods (e.g., Adam and RMSprop).

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
