(\boldsymbol{x}) - \nabla f(\boldsymbol{y})\| \leq L\|\boldsymbol{x} - \boldsymbol{y}\|$ for all $\boldsymbol{x}, \boldsymbol{y} \in \Theta$.*

**Assumption 4.** *The stochastic gradient has a finite second moment, i.e., there exists a constant $G > 0$ such that $\mathbb{E}[\|\boldsymbol{g}_t\|^2] \leq G^2$.*

In the literature on convergence analysis, it is common to analyze the convergence rate of $\min_t\{\mathbb{E}[\|\nabla f(\boldsymbol{\theta}_t))\|^2]\}$, where $\boldsymbol{\theta}_t$ represents the parameter value after $t$ parameter updates.

For the analysis of adaptive gradient methods (e.g., Adam and AMSGrad), many of previous works (Chen et al., 2019; Zhou et al., 2018; Défossez et al., 2022) make an additional assumption that the stochastic gradient $\boldsymbol{g}_t$ is uniformly bounded:

**Assumption 5.** *The stochastic gradient is uniformly upper-bounded, i.e., there exists a constant $G > 0$ such that $\|\boldsymbol{g}_t\| \leq G$.*

Note that when Assumption 5 holds, Assumption 4 is automatically satisfied. Therefore, Assumption 5 is a stronger assumption compared to Assumption 4. When we omit Assumption 5, it becomes challenging to analyze $\min_t\{\mathbb{E}[\|\nabla f(\boldsymbol{\theta}_t))\|^2]\}$ for adaptive gradient methods. As a result, the analysis often considers $\min_t\{\mathbb{E}[\|\nabla f(\boldsymbol{\theta}_t))\|^{4/3}]^{3/2}\}$ instead. In this paper, we focus on analyzing $\min_t\{\mathbb{E}[\|\nabla f(\boldsymbol{\theta}_t))\|^{4/3}]^{3/2}\}$, because one of our motivations is to address the omission of Assumption 5.

## 2.2 Review of Stochastic Optimization Algorithms for Nonconvex Objectives

The convergence of the vanilla SGD have been studied extensively in previous works. For smooth nonconvex functions, Ghadimi & Lan (2013) showed that SGD with a constant learning rate converges with an $\mathcal{O}(1/\sqrt{T})$ rate by setting $\alpha_t = \alpha = \Theta(1/\sqrt{T})$, where $\alpha_t$ is a learning rate at the $t$-th step, and $T$ is a total number of parameter updates. This convergence rate is known to be minimax optimal up to a constant (Drori & Shamir, 2020). For the diminishing learning rate scheme, the convergence bound of $\mathcal{O}(\log T/\sqrt{T})$ is well-known for $\alpha_t = \alpha/\sqrt{t}$ (Ghadimi & Lan, 2013). Recently, Wang et al. (2021) have proved that SGD with $\alpha_t = \alpha/\sqrt{t}$ can also achieve the optimal rate $\mathcal{O}(1/\sqrt{T})$ by additionally assuming that the objective $f$ is upper-bounded.

While the vanilla SGD is still one of the most popular choices for stochastic optimization, adaptive gradient methods are dominantly used especially for deep learning. In adaptive gradient methods, the parameter $\boldsymbol{\theta}$ is updated additionally using the second moment estimate $\boldsymbol{v}_t$ in the following form:

$$\boldsymbol{\theta}_t = \boldsymbol{\theta}_{t-1} - \alpha_t \frac{\boldsymbol{g}_t}{\sqrt{\boldsymbol{v}_t + \epsilon^2}}, \tag{2}$$

where $\epsilon$ is a small constant, the division between vectors is applied in an element-wise manner, and the addition between a vector $\boldsymbol{a}$ and a scalar $b$ is defined as $(\boldsymbol{a} + b)_i := a_i + b$. In AdaGrad (Duchi et al., 2011), $\boldsymbol{v}_t$ is defined as $\boldsymbol{v}_0 = \boldsymbol{0}$ and $\boldsymbol{v}_t = \boldsymbol{v}_{t-1} + \boldsymbol{g}_t \odot \boldsymbol{

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

| | Algorithm | Problem | Smoothness | Gradient Growth |
|---|---|---|---|---|
| Zhang et al. (2022) | Adam | Finite sum | $L$-Lipschitz $\boldsymbol{g}$ | $\mathbb{E}[\|g\|^2] \leq G_0^2 + G_1^2 \|\nabla f\|^2$ |
| Wang et al. (2022) | Adam | Finite sum | $(L_0, L_1)$-Lipschitz $\boldsymbol{g}$ | $\mathbb{E}[\|g\|^2] \leq G_0^2 + G_1^2 \|\nabla f\|^2$ |
| Li et al. (2023) | Adam | General | $(L_0, L_\rho)$-Lipschitz $\nabla f$ | $\|\boldsymbol{g} - \nabla f\| \leq \sigma$ |
| Défossez et al. (2022) | Adam | General | $L$-Lipschitz $\nabla f$ | $\mathbb{E}[\|\boldsymbol{g}\|^2] \leq G^2$ |
| Chen et al. (2019) | AMSGrad | General | $L$-Lipschitz $\nabla f$ | $\|\boldsymbol{g}\| \leq G$ |
| Zhou et al. (2018) | AMSGrad | General | $L$-Lipschitz $\nabla f$ | $\|\boldsymbol{g}\| \leq G$ |
| Ours | ADOPT | General | $L$-Lipschitz $\nabla f$ | $\mathbb{E}[\|\boldsymbol{g}\|^2] \leq G^2$ |

Table 3: Comparison of the problem settings between our analysis and other existing works.

| | Constraints | Convergence |
|---|---|---|
| Zhang et al. (2022) | $\beta_1 < \sqrt{\beta_2}, \beta_2 \geq \gamma(n)$ | $\mathcal{O}(\log T/\sqrt{T}) + \mathcal{O}(G_0)$ |
| Wang et al. (2022) | $\beta_1 < \sqrt{\beta_2}, \delta(\beta_2) = \mathcal{O}(1/G_1)$ | $\mathcal{O}(\log T/\sqrt{T}) + \mathcal{O}(G_0)$ |
| Li et al. (2023) | $\beta_1 < \sqrt{\beta_2}, \beta_1 \leq c(L_0, L_\rho, G)$ | $\mathcal{O}(1/\sqrt{T})$ |
| Défossez et al. (2022) | $\beta_1 < \sqrt{\beta_2}, 1 - \beta_2 = \Theta(1/T)$ | $\mathcal{O}(\log T/\sqrt{T})$ |
| Chen et al. (2019) | $\beta_1 < \sqrt{\beta_2}$ | $\mathcal{O}(\log T/\sqrt{T})$ |
| Zhou et al. (2018) | $\beta_1 < \sqrt{\beta_2}$ | $\mathcal{O}(1/\sqrt{T})$ |
| Ours | - | $\mathcal{O}(1/\sqrt{T})$ |

Table 4: Comparison of the convergence rate and imposed constraints on the hyperparameters between our analysis and other existing works. Please refer to the original papers for the definitions of $\gamma$ and $c$.

## A   DETAILED RELATIONSHIPS TO EXISTING ANALYSES

In this section, we discuss the relationships between our analysis and existing ones on the convergence of Adam-like optimizers in smooth nonconvex optimization problems. Tables 7 and 7 are a summary of comparisons between them in terms of their problem settings and derived convergence rates.

**Zhang et al. (2022)** focus on convergence of Adam in the finite sum problem, where the objective has a following form:

$$f(\boldsymbol{\theta}) = \sum_{i=1}^{n} f_i(\boldsymbol{\theta}). \tag{15}$$

$f_i$ is, for example, a loss function for $i$-th training sample. Although many deep learning problems can be formulated as a finite sum problem, training of the variational autoencoders (VAEs) or diffusion models is out of the finite-sum problem, since their objective is formulated as an infinite sum (i.e., an expectation over continuous variables). Moreover, they assume the stochastic gradient $\boldsymbol{g}$ is $L$-Lipschitz, whereas we only assume true gradient $\nabla f$ is $L$-Lipschitz. They also assume a growth condition as follows:

$$\mathbb{E}\left[\|\boldsymbol{g}_t\|^2\right] \leq G_0^2 + G_1^2 \|\nabla f(\boldsymbol{\theta}_{t-1})\|^2. \tag{16}$$

This growth condition is weaker than our Assumption 4. Assumption 4 is a special case of the growth condition where $G_1 = 0$. Their derived convergence rate has a constant factor of $\mathcal{O}(G_0)$; hence the strong growth condition (i.e., $G_0 = 0$) is required to assure convergence. Moreover, to achieve the convergence rate, one needs to choose sufficiently large $\beta_2$, which has to be tuned in a problem-dependent manner.

**Wang et al. (2022)** also focus on convergence of Adam in the finite sum problem, but they relax the $L$-Lipschitz condition on $\boldsymbol{g}$ to the $(L_0, L_1)$-Lipschitz condition. They also assume the growth condition in Eq. (16), and their convergence rate has the same order with Zhang et al. (2022), so it still requires the strong growth condition (i.e., $G_0 = 0$) to assure convergence. The condition of $\beta_2$ is also similar to Zhang et al. (2022).

**Li et al. (2023)** consider Adam's convergence on general smooth nonconvex problems. Similar to Wang et al. (2022), they use $(L_0, L_\rho)$-Lipschitz condition on the true gradient $\nabla f$. They also assume

that the gradient noise is almost surely bounded:

$$\|\boldsymbol{g} - \nabla f\| \le \sigma \tag{17}$$

The relationship between this assumption and our Assumption 4 is a little complicated. Assumption 4 is equivalent to a combination of the following two Assumptions:

**Assumption 6.** *The true gradient is uniformly bounded, i.e., there exist constants $G$ and $\sigma$ such that* $\|\nabla f(\boldsymbol{\theta})\|^2 \le G^2 - \sigma^2$ *and* $G > \sigma$.

**Assumption 7.** *Variance of the stochastic gradient is uniformly bounded, i.e., there exist a constant* $\sigma$ *such that* $\mathbb{E}\left[\|\boldsymbol{g}_t - \nabla f(\boldsymbol{\theta}_{t-1})\|^2\right] \le \sigma^2$.

The bounded noise assumption of Eq. (17) is strictly stronger than Assumption 7, but they do not assume the bounded true gradient (i.e., Assumption 6). The bounded noise assumption is often violated in practice (e.g., training of VAEs), because the gradient is often estimated using unbounded noise (i.e., Gaussian noise). Their convergence rate $\mathcal{O}(1/\sqrt{T})$ is better than Zhang et al. (2022) and Wang et al. (2022), while it still requires constraints on the hyperparameters, which have to be chosen in a problem-dependent manner.

**Défossez et al. (2022)** analyzes the convergence of Adam under exactly the same assumptions with ours, and they derive the $\mathcal{O}(\log T/\sqrt{T})$ rate, which is worse than our ADOPT's convergence rate. Moereover, to assure the convergence, $\beta_2$ has to be chosen dependently on the total number of iterations $T$.

**Chen et al. (2019)** and **Zhou et al. (2018)** analyze the convergence of AMSGrad for general smooth nonconvex problems, and derive the convergence rate of $\mathcal{O}(\log T/\sqrt{T})$ and $\mathcal{O}(1/\sqrt{T})$, respectively. However, to guarantee the convergence, the stochastic gradient $\boldsymbol{g}$ has to be bounded almost surely (Assumption 5), which is often violated in practice. In addition, the hyperparameter $\beta_1$ and $\beta_2$ should be chosen satisfying $\beta_1 < \sqrt{\beta_2}$. This constraint is relatively minor compared to the constraint imposed in the analyses of Adam, since it can be satisfied in a problem-independent manner.

## B  WITH-REPLACEMENT VS. WITHOUT-REPLACEMENT

In the optimization of finite-sum problems, practitioners often use *without-replacement sampling*, which is also known as *random shuffling*, to obtain stochastic gradient. In this case, the stochastic gradient has a small bias due to the lack of replacement, so Assumption 2 is violated. However, the vanilla SGD is known to converge with the without-replacement strategy (Haochen & Sra, 2019), and some of the analyses of Adam also adopt without-replacement sampling (Zhang et al., 2022; Wang et al., 2022).

Unfortunately, we find that our ADOPT has a counter example, in which ADOPT fails to converge when using without-replacement sampling. For example, when we consider minimizing $f(\theta) = \sum_{i=1}^{3} f_i(\theta)$, where $\theta \in [-1, 1]$, $f_1(\theta) = 1.9\theta$ and $f_2(\theta) = f_2(\theta) = -\theta$, it can be easily observed that ADOPT with $\beta_1 = \beta_2 = 0$ fails to converge to the correct solution, i.e., $\theta = 1$.

This non-convergence can be easily avoided by using the with-replacement strategy. Moreover, the difference between with- and without-replacement sampling becomes negligible when $n$ in the finite-sum $\sum_{i=1}^{n} f_i$ is large enough; hence it does not affect the practical performance very much. In fact, our experiments except for the toy example are performed using without-replacement sampling, but divergent behaviors are not observed. If one applies ADOPT to problems where the difference seems severe (e.g., when training with a small dataset), we recommend to use with-replacement sampling instead of random shuffling for stable training. When one uses PyTorch (Paszke et al., 2019) for the implementation, for example, with-replacement sampling can be easily applied by specifying `replacemnet=True` for `torch.utils.data.RandomSampler`, and feeding it to the `sampler` argument of `torch.utils.data.DataLoader`.

## C ON THE BIAS CORRECTION TECHNIQUE

A bias correction technique is originally proposed in Adam (Kingma & Ba, 2014), in which the momentum and the second moment estimate are scaled as follows:

$$\tilde{\boldsymbol{m}}_t = \boldsymbol{m}_t / \left(1 - \beta_1^t\right) \tag{18}$$

$$\tilde{\boldsymbol{v}}_t = \boldsymbol{v}_t / \left(1 - \beta_2^t\right) \tag{19}$$

This technique prevents $\boldsymbol{m}_t$ and $\boldsymbol{v}_t$ from being biased to the initial value (i.e., $\boldsymbol{0}$ for Adam), and contributes to stable training of Adam. Basically, this technique can be directly applicable to our ADOPT, but we empirically find that ADOPT works adequately well even when the bias correction is not applied. Our hypothesis about the reason of it is that $\boldsymbol{v}_t$ is initialized to $\boldsymbol{1}$ for ADOPT to prevent it from being too small at the first parameter update, so the effect of bias correction is limited compared to Adam. Moreover, as for the momentum $\boldsymbol{m}_t$, initialization with $\boldsymbol{0}$ works like a warm-up of the learning rate, so it may have a positive effect for stable training especially in the early phase of optimization. Although there might be a better way to introduce a technique like the bias correction for ADOPT, we leave it for future work.

## D RECOMMENDATION OF HYPERPARAMETER SETTINGS FOR ADOPT

We experimentally find that our ADOPT works similarly to Adam when the same hyperparameters are used, but $\epsilon$ should be set to a little larger value (e.g., $1 \times 10^{-6}$) for ADOPT compared to Adam, in which $\epsilon$ is set to $1 \times 10^{-8}$ by default. Our recommendation of the hyperparameter settings for ADOPT is provided in Table 5.

| | |
|---|---|
| $\beta_1$ | 0.9 |
| $\beta_2$ | 0.999 |
| $\epsilon$ | $1 \times 10^{-6}$ |

Table 5: Recommended hyperparameters for the ADOPT algorithm

## E THEOREMS

**Theorem 3.** *Under Assumptions 1, 2, 3, and 4, if the objective $f$ is upper-bounded by $f_{\mathrm{sup}}$, the following holds for the ADOPT algorithm with a learning rate $\alpha_t = \alpha/\sqrt{t}$:*

$$
\begin{aligned}
\min_{t=1,\ldots,T} &\left\{ \mathbb{E}\left[\|\nabla f\left(\boldsymbol{\theta}_t\right)\|^{4/3}\right]^{3/2} \right\} \\
&\leq \frac{3\sqrt{\max\{G^2,1\} + \epsilon^2}}{2\left((T+1)^{3/2} - 1\right)} \left( \frac{f_{\mathrm{sup}} - f_{\mathrm{inf}}}{\alpha}(T+1) + \left(\frac{\sqrt{2}\alpha\beta_1 G^2 L}{\epsilon^2(1-\beta_1)} + \frac{\alpha G^2 L}{2\epsilon^2}\right)T \right) \\
&+ \frac{3\sqrt{\max\{G^2,1\} + \epsilon^2}}{2\left((T+1)^{3/2} - 1\right)} \left( \frac{2\sqrt{2}\beta_1 G^2}{\epsilon(1-\beta_1)}\left(\sqrt{T+1} - 1\right) + \frac{\alpha\beta_1^2 G^2 L}{\epsilon(1-\beta_1)^2}\frac{T}{T+1} \right) \\
&+ \frac{3\sqrt{\max\{G^2,1\} + \epsilon^2}}{2\left((T+1)^{3/2} - 1\right)} \left( \frac{2\alpha\beta_1^2 G^2 L}{\epsilon(1-\beta_1)^2} + \frac{\alpha^2\beta_1 G^2 L}{\sqrt{2}\epsilon^2(1-\beta_1)} \right)\log(T+1).
\end{aligned}
\tag{20}
$$

## F PROOFS

*Proof of Theorems 2 and 3.* We define $\phi_t$ for $t \geq 1$ as follows:

$$\phi_t = \frac{1}{1-\beta_1}\boldsymbol{\theta}_t - \frac{\beta_1}{1-\beta_1}\boldsymbol{\theta}_{t-1}. \tag{21}$$

We also define $\phi_0 = \theta_0$. By Assumption 3, the following holds for $t \geq 1$:

$$f\left(\phi_t\right) \leq f\left(\phi_{t-1}\right) + \nabla f\left(\phi_{t-1}\right)^\top \left(\phi_t - \phi_{t-1}\right) + \frac{L}{2} \left\|\phi_t - \phi_{t-1}\right\|^2 \tag{22}$$

$$= f\left(\phi_{t-1}\right) + \nabla f\left(\theta_{t-1}\right)^\top \left(\phi_t - \phi_{t-1}\right)$$
$$+ \left(\nabla f\left(\phi_{t-1}\right) - \nabla f\left(\theta_{t-1}\right)\right)^\top \left(\phi_t - \phi_{t-1}\right) + \frac{L}{2} \left\|\phi_t - \phi_{t-1}\right\|^2 \tag{23}$$

$$\leq f\left(\phi_{t-1}\right) + \nabla f\left(\theta_{t-1}\right)^\top \left(\phi_t - \phi_{t-1}\right)$$
$$+ \left\|\nabla f\left(\phi_{t-1}\right) - \nabla f\left(\theta_{t-1}\right)\right\| \left\|\phi_t - \phi_{t-1}\right\| + \frac{L}{2} \left\|\phi_t - \phi_{t-1}\right\|^2 \tag{24}$$

$$\leq f\left(\phi_{t-1}\right) + \nabla f\left(\theta_{t-1}\right)^\top \left(\phi_t - \phi_{t-1}\right)$$
$$+ L \left\|\phi_{t-1} - \theta_{t-1}\right\| \left\|\phi_t - \phi_{t-1}\right\| + \frac{L}{2} \left\|\phi_t - \phi_{t-1}\right\|^2, \tag{25}$$

where the second inequality holds due to the Cauchy-Schwarz inequality, and the last inequality holds due to Assumption 3.

By taking the expectation, the following holds:

$$\mathbb{E}\left[f\left(\phi_t\right)\right] \leq \mathbb{E}\left[f\left(\phi_{t-1}\right)\right] + \mathbb{E}\left[\nabla f\left(\theta_{t-1}\right)^\top \left(\phi_t - \phi_{t-1}\right)\right]$$
$$+ L\mathbb{E}\left[\left\|\phi_{t-1} - \theta_{t-1}\right\| \left\|\phi_t - \phi_{t-1}\right\|\right] + \frac{L}{2}\mathbb{E}\left[\left\|\phi_t - \phi_{t-1}\right\|^2\right] \tag{26}$$

$$\leq \mathbb{E}\left[f\left(\phi_{t-1}\right)\right] + \frac{\left(\alpha_{t-1} - \alpha_t\right)\beta_1\left(1 - \beta_1^{t-1}\right)G^2}{\left(1 - \beta_1\right)\sqrt{\beta_2^{t-2} + \epsilon^2}}$$

$$- \alpha_t \frac{\mathbb{E}\left[\left\|\nabla f\left(\theta_{t-1}\right)\right\|_i^{4/3}\right]^{3/2}}{\sqrt{\max\left\{G^2 + (1 - G^2)\beta_2^T, 1\right\} + \epsilon^2}} \tag{27}$$

$$+ \frac{\alpha_{t-1}\left(\alpha_{t-1} - \alpha_t\right)\beta_1^2\left(1 - \beta_1^{t-1}\right)G^2 L}{\left(\beta_2^{t-2} + \epsilon^2\right)\left(1 - \beta_1\right)^2} + \frac{\alpha_t\alpha_{t-1}\beta_1\sqrt{1 - \beta_1^{t-1}}G^2 L}{\left(1 - \beta_1\right)\sqrt{\beta_2^{t-1} + \epsilon^2}\sqrt{\beta_2^{t-2} + \epsilon^2}}$$

$$+ \frac{\left(\alpha_{t-1} - \alpha_t\right)^2\beta_1^2\left(1 - \beta_1^{t-1}\right)G^2 L}{2\left(1 - \beta_1\right)^2\left(\beta_2^{t-2} + \epsilon^2\right)} + \frac{\alpha_t^2 G^2 L}{2\left(\beta_2^{t-1} + \epsilon^2\right)}$$

$$+ \frac{\alpha_t\left(\alpha_{t-1} - \alpha_t\right)\beta_1\sqrt{1 - \beta_1^{t-1}}G^2 L}{2\left(1 - \beta_1\right)\sqrt{\beta_2^{t-1} + \epsilon^2}\sqrt{\beta_2^{t-2} + \epsilon^2}}. \tag{28}$$

When $\alpha_t = \alpha$, the following holds:

$$\mathbb{E}\left[f\left(\phi_t\right)\right] \leq \mathbb{E}\left[f\left(\phi_{t-1}\right)\right] - \alpha\frac{\mathbb{E}\left[\left\|\nabla f\left(\theta_{t-1}\right)\right\|_i^{4/3}\right]^{3/2}}{\sqrt{\max\left\{G^2 + (1 - G^2)\beta_2^T, 1\right\} + \epsilon^2}}$$

$$+ \frac{\alpha^2\beta_1\sqrt{1 - \beta_1^{t-1}}G^2 L}{\left(1 - \beta_1\right)\sqrt{\beta_2^{t-1} + \epsilon^2}\sqrt{\beta_2^{t-2} + \epsilon^2}} + \frac{\alpha^2 G^2 L}{2\left(\beta_2^{t-1} + \epsilon^2\right)} \tag{29}$$

$$\leq \mathbb{E}\left[f\left(\phi_{t-1}\right)\right] - \alpha\frac{\mathbb{E}\left[\left\|\nabla f\left(\theta_{t-1}\right)\right\|^{4/3}\right]^{3/2}}{\sqrt{\max\left\{G^2 + (1 - G^2)\beta_2^T, 1\right\} + \epsilon^2}}$$

$$+ \frac{\alpha^2\left(1 + \beta_1\right)G^2 L}{2\left(1 - \beta_1\right)\left(\beta_2^{t-1} + \epsilon^2\right)}. \tag{30}$$

Telescoping it for $t = 1, \ldots, T$, we have

$$\mathbb{E}\left[f\left(\boldsymbol{\phi}_T\right)\right] \leq f\left(\boldsymbol{\theta}_0\right) - \alpha \frac{\sum_{t=1}^{T} \mathbb{E}\left[\|\nabla f\left(\boldsymbol{\theta}_{t-1}\right)\|^{4/3}\right]^{3/2}}{\sqrt{\max\left\{G^2 + (1 - G^2)\beta_2^T, 1\right\} + \epsilon^2}} + \frac{\alpha^2\left(1 + \beta_1\right)G^2 L}{2\left(1 - \beta_1\right)} \sum_{t=0}^{T-1} \frac{1}{\beta_2^t + \epsilon^2} \tag{31}$$

$$\leq f\left(\boldsymbol{\theta}_0\right) - \alpha \frac{\sum_{t=1}^{T} \mathbb{E}\left[\|\nabla f\left(\boldsymbol{\theta}_{t-1}\right)\|^{4/3}\right]^{3/2}}{\sqrt{\max\left\{G^2 + (1 - G^2)\beta_2^T, 1\right\} + \epsilon^2}} + \frac{\alpha^2\left(1 + \beta_1\right)G^2 L}{2\left(1 - \beta_1\right)} \int_0^T \frac{1}{\beta_2^t + \epsilon^2} dt \tag{32}$$

$$\leq f\left(\boldsymbol{\theta}_0\right) - \alpha \frac{\sum_{t=1}^{T} \mathbb{E}\left[\|\nabla f\left(\boldsymbol{\theta}_{t-1}\right)\|^{4/3}\right]^{3/2}}{\sqrt{\max\left\{G^2 + (1 - G^2)\beta_2^T, 1\right\} + \epsilon^2}}$$
$$+ \frac{\alpha^2\left(1 + \beta_1\right)G^2 L}{2\left(1 - \beta_1\right)\epsilon^2}\left(T - \frac{1}{\log\beta_2}\log\left(\frac{\beta_2^T + \epsilon^2}{1 + \epsilon^2}\right)\right) \tag{33}$$

By rearranging the terms, we have

$$\min_{t=1,\ldots,T}\left\{\mathbb{E}\left[\|\nabla f\left(\boldsymbol{\theta}_{t-1}\right)\|^{4/3}\right]^{3/2}\right\}$$

$$\leq \frac{\sum_{t=1}^{T} \mathbb{E}\left[\|\nabla f\left(\boldsymbol{\theta}_{t-1}\right)\|^{4/3}\right]^{3/2}}{T} \tag{34}$$

$$\leq \sqrt{\max\left\{G^2, 1\right\} + \epsilon^2}\left(\frac{f\left(\boldsymbol{\theta}_0\right) - f_{\inf}}{\alpha T} + \alpha C_2\left(1 - \frac{1}{T\log\beta_2}\log\left(\frac{\beta_2^T + \epsilon^2}{1 + \epsilon^2}\right)\right)\right) \tag{35}$$

$$\leq C_1\left(T\right)\left(\frac{f\left(\boldsymbol{\theta}_0\right) - f_{\inf}}{\alpha T} + \alpha C_2\left(1 - \frac{1}{T\log\beta_2}\log\left(\frac{\beta_2^T + \epsilon^2}{1 + \epsilon^2}\right)\right)\right), \tag{36}$$

where $C_1\left(T\right) = \sqrt{\max\left\{G^2 + (1 - G^2)\beta_2^T, 1\right\} + \epsilon^2}$, $C_2 = \frac{(1 + \beta_1)G^2 L}{2(1 - \beta_1)\epsilon^2}$.

When $\alpha_t = \alpha/\sqrt{t}$, the following holds for $t \geq 2$:

$$\alpha_{t-1} - \alpha_t = \alpha\left(\frac{1}{\sqrt{t-1}} - \frac{1}{\sqrt{t}}\right) \tag{37}$$

$$= \frac{\alpha\left(\sqrt{t} - \sqrt{t-1}\right)}{\sqrt{t(t-1)}} \tag{38}$$

$$= \frac{\alpha}{\sqrt{t(t-1)}\left(\sqrt{t} + \sqrt{t-1}\right)} \tag{39}$$

$$\leq \frac{\alpha}{2(t-1)^{3/2}} \tag{40}$$

$$\leq \frac{\sqrt{2}\alpha}{t^{3/2}}. \tag{41}$$

This also holds for $t = 1$ by defining $\alpha_0 = \alpha$. Applying it to Eq. (28), we have

$$
\mathbb{E}\left[f\left(\phi_t\right)\right] \le \mathbb{E}\left[f\left(\phi_{t-1}\right)\right] + \frac{\left(\alpha_{t-1} - \alpha_t\right)\beta_1\left(1 - \beta_1^{t-1}\right)G^2}{\left(1 - \beta_1\right)\sqrt{\beta_2^{t-2} + \epsilon^2}}
$$

$$
- \alpha_t \frac{\mathbb{E}\left[\left\|\nabla f\left(\boldsymbol{\theta}_{t-1}\right)\right\|_i^{4/3}\right]^{3/2}}{\sqrt{\max\left\{G^2 + (1 - G^2)\beta_2^T, 1\right\} + \epsilon^2}}
$$

$$
+ \frac{\alpha_{t-1}\left(\alpha_{t-1} - \alpha_t\right)\beta_1^2\left(1 - \beta_1^{t-1}\right)G^2 L}{\left(\beta_2^{t-2} + \epsilon^2\right)\left(1 - \beta_1\right)^2} + \frac{\alpha_t\alpha_{t-1}\beta_1\sqrt{1 - \beta_1^{t-1}}G^2 L}{\left(1 - \beta_1\right)\sqrt{\beta_2^{t-1} + \epsilon^2}\sqrt{\beta_2^{t-2} + \epsilon^2}}
$$

$$
+ \frac{\left(\alpha_{t-1} - \alpha_t\right)^2\beta_1^2\left(1 - \beta_1^{t-1}\right)G^2 L}{2\left(1 - \beta_1\right)^2\left(\beta_2^{t-2} + \epsilon^2\right)} + \frac{\alpha_t^2 G^2 L}{2\left(\beta_2^{t-1} + \epsilon^2\right)}
$$

$$
+ \frac{\alpha_t\left(\alpha_{t-1} - \alpha_t\right)\beta_1\sqrt{1 - \beta_1^{t-1}}G^2 L}{2\left(1 - \beta_1\right)\sqrt{\beta_2^{t-1} + \epsilon^2}\sqrt{\beta_2^{t-2} + \epsilon^2}} \tag{42}
$$

$$
\le \mathbb{E}\left[f\left(\phi_{t-1}\right)\right] + \frac{\sqrt{2}\alpha\beta_1\left(1 - \beta_1^{t-1}\right)G^2}{t^{3/2}\left(1 - \beta_1\right)\sqrt{\beta_2^{t-2} + \epsilon^2}}
$$

$$
- \frac{\alpha}{\sqrt{t}}\frac{\mathbb{E}\left[\left\|\nabla f\left(\boldsymbol{\theta}_{t-1}\right)\right\|_i^{4/3}\right]^{3/2}}{\sqrt{\max\left\{G^2 + (1 - G^2)\beta_2^T, 1\right\} + \epsilon^2}}
$$

$$
+ \frac{2\alpha^2\beta_1^2 G^2 L}{\left(\beta_2^{t-1} + \epsilon^2\right)\left(1 - \beta_1\right)^2 t^2} + \frac{\sqrt{2}\alpha^2\beta_1 G^2 L}{\left(1 - \beta_1\right)\left(\beta_2^{t-1} + \epsilon^2\right)t}
$$

$$
+ \frac{\alpha^2\beta_1^2 G^2 L}{\left(1 - \beta_1\right)^2\left(\beta_2^{t-1} + \epsilon^2\right)t^3} + \frac{\alpha^2 G^2 L}{2\left(\beta_2^{t-1} + \epsilon^2\right)t}
$$

$$
+ \frac{\alpha^2\beta_1 G^2 L}{\sqrt{2}\left(1 - \beta_1\right)\left(\beta_2^{t-1} + \epsilon^2\right)t^2} \tag{43}
$$

$$
= \mathbb{E}\left[f\left(\phi_{t-1}\right)\right] - \frac{\alpha}{\sqrt{t}}\frac{\mathbb{E}\left[\left\|\nabla f\left(\boldsymbol{\theta}_{t-1}\right)\right\|_i^{4/3}\right]^{3/2}}{\sqrt{\max\left\{G^2 + (1 - G^2)\beta_2^T, 1\right\} + \epsilon^2}}
$$

$$
+ \frac{\alpha^2\left(1 + \left(2\sqrt{2} - 1\right)\beta_1\right)G^2 L}{2\left(1 - \beta_1\right)\left(\beta_2^{t-1} + \epsilon^2\right)} \cdot t^{-1} + \frac{\sqrt{2}\alpha\beta_1 G^2}{\left(1 - \beta_1\right)\sqrt{\beta_2^T + \epsilon^2}} \cdot t^{-\frac{3}{2}}
$$

$$
+ \frac{\alpha^2\beta_1\left(1 + \left(2\sqrt{2} - 1\right)\beta_1\right)G^2 L}{\sqrt{2}\left(1 - \beta\right)^2\left(\beta_2^T + \epsilon^2\right)} \cdot t^{-2} + \frac{\alpha^2\beta_1^2 G^2 L}{\left(1 - \beta_1\right)^2\left(\beta_2^T + \epsilon^2\right)} \cdot t^{-3}. \tag{44}
$$

Multiplying $t$ to the both sides and rearranging the terms, we have

$$
\frac{\sqrt{t}\,\mathbb{E}\left[\left\|\nabla f\left(\boldsymbol{\theta}_{t-1}\right)\right\|^{4/3}\right]^{3/2}}{\sqrt{\max\left\{G^2, 1\right\} + \epsilon^2}}
$$

$$
\le \frac{\mathbb{E}\left[f\left(\phi_{t-1}\right) - f\left(\phi_t\right)\right]}{\alpha} \cdot t + \frac{\alpha\left(1 + \left(2\sqrt{2} - 1\right)\beta_1\right)G^2 L}{2\left(1 - \beta_1\right)\left(\beta_2^{t-1} + \epsilon^2\right)} + \frac{\sqrt{2}\beta_1 G^2}{\left(1 - \beta_1\right)\sqrt{\beta_2^T + \epsilon^2}} \cdot t^{-\frac{1}{2}} \tag{45}
$$

$$
+ \frac{\alpha\beta_1\left(1 + \left(2\sqrt{2} - 1\right)\beta_1\right)G^2 L}{\sqrt{2}\left(1 - \beta\right)^2\left(\beta_2^T + \epsilon^2\right)}t^{-1} + \frac{\alpha\beta_1^2 G^2 L}{\left(1 - \beta_1\right)^2\left(\beta_2^T + \epsilon^2\right)}t^{-2} \tag{46}
$$

$$\sum_{t=1}^{T} \frac{\sqrt{t}\, \mathbb{E}\left[\|\nabla f\left(\boldsymbol{\theta}_{t-1}\right)\|^{4/3}\right]^{3/2}}{\sqrt{\max\left\{G^2 + (1-G^2)\beta_2^T, 1\right\} + \epsilon^2}}$$

$$\leq \frac{f\left(\boldsymbol{\phi}_0\right) - T f\left(\boldsymbol{\phi}_T\right) + \sum_{t=1}^{T-1} f\left(\boldsymbol{\phi}_t\right)}{\alpha} + \frac{\alpha\left(1 + \left(2\sqrt{2}-1\right)\beta_1\right) G^2 L}{2\left(1-\beta_1\right)} \sum_{t=1}^{T} \frac{1}{\beta_2^{t-1} + \epsilon^2}$$

$$+ \frac{\sqrt{2}\beta_1 G^2}{\left(1-\beta_1\right)\sqrt{\beta_2^T + \epsilon^2}} \sum_{t=1}^{T} t^{-\frac{1}{2}} + \frac{\alpha\beta_1\left(1 + \left(2\sqrt{2}-1\right)\beta_1\right) G^2 L}{\sqrt{2}\left(1-\beta\right)^2\left(\beta_2^T + \epsilon^2\right)} \sum_{t=1}^{T} t^{-1}$$

$$+ \frac{\alpha\beta_1^2 G^2 L}{\left(1-\beta_1\right)^2\left(\beta_2^T + \epsilon^2\right)} \sum_{t=1}^{T} t^{-2} \tag{47}$$

$$\leq \frac{f_{\text{sup}} - f_{\text{inf}}}{\alpha} T + \frac{\alpha\left(1 + \left(2\sqrt{2}-1\right)\beta_1\right) G^2 L}{2\left(1-\beta_1\right)} \int_0^T \frac{1}{\beta_2^t + \epsilon^2}\, dt$$

$$+ \frac{\sqrt{2}\beta_1 G^2}{\left(1-\beta_1\right)\sqrt{\beta_2^T + \epsilon^2}} \left(1 + \int_1^T t^{-\frac{1}{2}}\, dt\right) + \frac{\alpha\beta_1\left(1 + \left(2\sqrt{2}-1\right)\beta_1\right) G^2 L}{\sqrt{2}\left(1-\beta\right)^2\left(\beta_2^T + \epsilon^2\right)} \left(1 + \int_1^T t^{-1}\, dt\right)$$

$$+ \frac{\alpha\beta_1^2 G^2 L}{\left(1-\beta_1\right)^2\left(\beta_2^T + \epsilon^2\right)} \left(1 + \int_1^T t^{-2}\, dt\right) \tag{48}$$

$$\leq \frac{f_{\text{sup}} - f_{\text{inf}}}{\alpha} T + \frac{\alpha\left(1 + \left(2\sqrt{2}-1\right)\beta_1\right) G^2 L}{2\left(1-\beta_1\right)\epsilon^2} \left(T - \frac{1}{\log\beta_2}\log\left(\frac{\beta_2^T + \epsilon^2}{1 + \epsilon^2}\right)\right)$$

$$+ \frac{\sqrt{2}\beta_1 G^2}{\left(1-\beta_1\right)\sqrt{\beta_2^T + \epsilon^2}} \left(2\sqrt{T} - 1\right) + \frac{\alpha\beta_1\left(1 + \left(2\sqrt{2}-1\right)\beta_1\right) G^2 L}{\sqrt{2}\left(1-\beta\right)^2\left(\beta_2^T + \epsilon^2\right)} \left(1 + \log T\right)$$

$$+ \frac{\alpha\beta_1^2 G^2 L}{\left(1-\beta_1\right)^2\left(\beta_2^T + \epsilon^2\right)} \left(2 - \frac{1}{T}\right) \tag{49}$$

Therefore, the following bound is derived.

$$\min_{t=1,\ldots,T}\left\{\mathbb{E}\left[\|\nabla f\left(\boldsymbol{\theta}_{t-1}\right)\|^{4/3}\right]^{3/2}\right\}$$

$$\leq \frac{\sum_{t=1}^{T}\sqrt{t}\, \mathbb{E}\left[\|\nabla f\left(\boldsymbol{\theta}_{t-1}\right)\|^{4/3}\right]^{3/2}}{\sum_{t=1}^{T}\sqrt{t}} \tag{50}$$

$$\leq \frac{\sum_{t=1}^{T}\sqrt{t}\, \mathbb{E}\left[\|\nabla f\left(\boldsymbol{\theta}_{t-1}\right)\|^{4/3}\right]^{3/2}}{\int_0^T\sqrt{t}\, dt} \tag{51}$$

$$\leq \frac{3 C_T\left(f_{\text{sup}} - f_{\text{inf}}\right)}{2\alpha}\frac{1}{\sqrt{T}} + \frac{3\alpha\left(1 + \left(2\sqrt{2}-1\right)\beta_1\right) C_T G^2 L}{4\left(1-\beta_1\right)\epsilon^2} \left(\frac{1}{\sqrt{T}} - \frac{1}{T^{3/2}\log\beta_2}\log\left(\frac{\beta_2^T + \epsilon^2}{1 + \epsilon^2}\right)\right)$$

$$+ \frac{3\beta_1 C_T G^2}{\sqrt{2}\left(1-\beta_1\right)\sqrt{\beta_2^T + \epsilon^2}} \left(\frac{2}{T} - \frac{1}{T^{3/2}}\right)$$

$$+ \frac{3\alpha\beta_1\left(1 + \left(2\sqrt{2}-1\right)\beta_1\right) C_T G^2 L}{2\sqrt{2}\left(1-\beta\right)^2\left(\beta_2^T + \epsilon^2\right)} \left(\frac{1}{T^{3/2}} + \frac{\log T}{T^{3/2}}\right)$$

$$+ \frac{3\alpha\beta_1^2 C_T G^2 L}{2\left(1-\beta_1\right)^2\left(\beta_2^T + \epsilon^2\right)} \left(\frac{2}{T^{3/2}} - \frac{1}{T^{5/2}}\right), \tag{52}$$

where $C_T = \sqrt{\max\left\{G^2 + (1-G^2)\beta_2^T, 1\right\} + \epsilon^2}$.

$$\mathbb{E}\left[f\left(\boldsymbol{\phi}_t\right)\right] \leq \mathbb{E}\left[f\left(\boldsymbol{\phi}_{t-1}\right)\right] + \frac{\left(\alpha_{t-1} - \alpha_t\right)\beta_1 G^2}{\epsilon\left(1-\beta_1\right)} - \alpha_t \frac{\mathbb{E}\left[\|\nabla f\left(\boldsymbol{\theta}_{t-1}\right)\|^{4/3}\right]^{3/2}}{\sqrt{\max\left\{G^2,1\right\} + \epsilon^2}} \tag{53}$$

$$+ \frac{\alpha_{t-1}\left(\alpha_{t-1} - \alpha_t\right)\beta_1^2 G^2 L}{\epsilon\left(1-\beta_1\right)^2} + \frac{\alpha_t \alpha_{t-1}\beta_1 G^2 L}{\epsilon^2\left(1-\beta_1\right)}$$

$$+ \frac{\left(\alpha_{t-1} - \alpha_t\right)^2 \beta_1^2 G^2 L}{2\epsilon\left(1-\beta_1\right)^2} + \frac{\alpha_t^2 G^2 L}{2\epsilon^2} + \frac{\alpha_t\left(\alpha_{t-1} - \alpha_t\right)\beta_1 G^2 L}{2\epsilon^2\left(1-\beta_1\right)} \tag{54}$$

$$\leq \mathbb{E}\left[f\left(\boldsymbol{\phi}_{t-1}\right)\right] + \frac{\sqrt{2}\alpha\beta_1 G^2}{\epsilon\left(1-\beta_1\right)t^{3/2}} - \frac{\alpha}{\sqrt{t}}\frac{\mathbb{E}\left[\|\nabla f\left(\boldsymbol{\theta}_{t-1}\right)\|^{4/3}\right]^{3/2}}{\sqrt{\max\left\{G^2,1\right\} + \epsilon^2}} \tag{55}$$

$$+ \frac{2\alpha^2\beta_1^2 G^2 L}{\epsilon\left(1-\beta_1\right)^2 t^2} + \frac{\sqrt{2}\alpha^2\beta_1 G^2 L}{\epsilon^2\left(1-\beta_1\right)t}$$

$$+ \frac{\alpha^2\beta_1^2 G^2 L}{\epsilon\left(1-\beta_1\right)^2 t^3} + \frac{\alpha^2 G^2 L}{2\epsilon^2 t} + \frac{\alpha^2\beta_1 G^2 L}{\sqrt{2}\epsilon^2\left(1-\beta_1\right)t^2}. \tag{56}$$

Multiplying $t$ to the both sides and rearranging the terms, we have

$$\frac{\sqrt{t}\,\mathbb{E}\left[\|\nabla f\left(\boldsymbol{\theta}_{t-1}\right)\|^{4/3}\right]^{3/2}}{\sqrt{\max\left\{G^2,1\right\} + \epsilon^2}}$$

$$\leq \frac{\mathbb{E}\left[f\left(\boldsymbol{\phi}_{t-1}\right) - f\left(\boldsymbol{\phi}_t\right)\right]}{\alpha}t + \frac{\sqrt{2}\alpha\beta_1 G^2 L}{\epsilon^2\left(1-\beta_1\right)} + \frac{\alpha G^2 L}{2\epsilon^2} \tag{57}$$

$$+ \frac{\sqrt{2}\beta_1 G^2}{\epsilon\left(1-\beta_1\right)}t^{-\frac{1}{2}} + \left(\frac{2\alpha\beta_1^2 G^2 L}{\epsilon\left(1-\beta_1\right)^2} + \frac{\alpha^2\beta_1 G^2 L}{\sqrt{2}\epsilon^2\left(1-\beta_1\right)}\right)t^{-1} + \frac{\alpha\beta_1^2 G^2 L}{\epsilon\left(1-\beta_1\right)^2}t^{-2} \tag{58}$$

Telescoping it for $t = 2, \ldots, T+1$, we have

$$\sum_{t=1}^{T}\frac{\sqrt{t+1}\,\mathbb{E}\left[\|\nabla f\left(\boldsymbol{\theta}_t\right)\|^{4/3}\right]^{3/2}}{\sqrt{\max\left\{G^2,1\right\} + \epsilon^2}}$$

$$\leq \frac{2f\left(\boldsymbol{\phi}_1\right) - \left(T+1\right)f\left(\boldsymbol{\phi}_{T+1}\right) + \sum_{t=2}^{T}f\left(\boldsymbol{\phi}_t\right)}{\alpha} + \left(\frac{\sqrt{2}\alpha\beta_1 G^2 L}{\epsilon^2\left(1-\beta_1\right)} + \frac{\alpha G^2 L}{2\epsilon^2}\right)T \tag{59}$$

$$+ \frac{\sqrt{2}\beta_1 G^2}{\epsilon\left(1-\beta_1\right)}\sum_{t=2}^{T+1}t^{-\frac{1}{2}} + \left(\frac{2\alpha\beta_1^2 G^2 L}{\epsilon\left(1-\beta_1\right)^2} + \frac{\alpha^2\beta_1 G^2 L}{\sqrt{2}\epsilon^2\left(1-\beta_1\right)}\right)\sum_{t=2}^{T+1}t^{-1} + \frac{\alpha\beta_1^2 G^2 L}{\epsilon\left(1-\beta_1\right)^2}\sum_{t=2}^{T+1}t^{-2}$$

$$\leq \frac{f_{\sup} - f_{\inf}}{\alpha}\left(T+1\right) + \left(\frac{\sqrt{2}\alpha\beta_1 G^2 L}{\epsilon^2\left(1-\beta_1\right)} + \frac{\alpha G^2 L}{2\epsilon^2}\right)T + \frac{\sqrt{2}\beta_1 G^2}{\epsilon\left(1-\beta_1\right)}\int_1^{T+1}t^{-\frac{1}{2}}dt \tag{60}$$

$$+ \left(\frac{2\alpha\beta_1^2 G^2 L}{\epsilon\left(1-\beta_1\right)^2} + \frac{\alpha^2\beta_1 G^2 L}{\sqrt{2}\epsilon^2\left(1-\beta_1\right)}\right)\int_1^{T+1}t^{-1}dt + \frac{\alpha\beta_1^2 G^2 L}{\epsilon\left(1-\beta_1\right)^2}\int_1^{T+1}t^{-2}dt$$

$$\leq \frac{f_{\sup} - f_{\inf}}{\alpha}\left(T+1\right) + \left(\frac{\sqrt{2}\alpha\beta_1 G^2 L}{\epsilon^2\left(1-\beta_1\right)} + \frac{\alpha G^2 L}{2\epsilon^2}\right)T + \frac{2\sqrt{2}\beta_1 G^2}{\epsilon\left(1-\beta_1\right)}\left(\sqrt{T+1}-1\right)$$

$$+ \left(\frac{2\alpha\beta_1^2 G^2 L}{\epsilon\left(1-\beta_1\right)^2} + \frac{\alpha^2\beta_1 G^2 L}{\sqrt{2}\epsilon^2\left(1-\beta_1\right)}\right)\log\left(T+1\right) + \frac{\alpha\beta_1^2 G^2 L}{\epsilon\left(1-\beta_1\right)^2}\frac{T}{T+1}. \tag{61}$$

Therefore, the following bound is derived.

$$
\min_{t=1,\ldots,T} \left\{ \mathbb{E}\left[ \|\nabla f(\boldsymbol{\theta}_t)\|^{4/3} \right]^{3/2} \right\}
$$

$$
\leq \frac{\sum_{t=1}^T \sqrt{t+1}\, \mathbb{E}\left[ \|\nabla f(\boldsymbol{\theta}_t)\|^{4/3} \right]^{3/2}}{\sum_{t=1}^T \sqrt{t+1}} \tag{62}
$$

$$
\leq \frac{\sum_{t=1}^T \sqrt{t+1}\, \mathbb{E}\left[ \|\nabla f(\boldsymbol{\theta}_t)\|^{4/3} \right]^{3/2}}{\int_0^T \sqrt{t+1}\, dt} \tag{63}
$$

$$
\leq \frac{3\sqrt{\max\{G^2,1\}+\epsilon^2}}{2\left((T+1)^{3/2}-1\right)} \left( \frac{f_{\sup}-f_{\inf}}{\alpha}(T+1) + \left( \frac{\sqrt{2}\alpha\beta_1 G^2 L}{\epsilon^2(1-\beta_1)} + \frac{\alpha G^2 L}{2\epsilon^2} \right) T \right)
$$

$$
+ \frac{3\sqrt{\max\{G^2,1\}+\epsilon^2}}{2\left((T+1)^{3/2}-1\right)} \left( \frac{2\sqrt{2}\beta_1 G^2}{\epsilon(1-\beta_1)}\left(\sqrt{T+1}-1\right) + \frac{\alpha\beta_1^2 G^2 L}{\epsilon(1-\beta_1)^2}\frac{T}{T+1} \right)
$$

$$
+ \frac{3\sqrt{\max\{G^2,1\}+\epsilon^2}}{2\left((T+1)^{3/2}-1\right)} \left( \frac{2\alpha\beta_1^2 G^2 L}{\epsilon(1-\beta_1)^2} + \frac{\alpha^2\beta_1 G^2 L}{\sqrt{2}\epsilon^2(1-\beta_1)} \right) \log(T+1). \tag{64}
$$

$\square$

## G  LEMMAS

**Lemma 4.** *For all $\boldsymbol{\theta} \in \mathbb{R}^D$ and $t \geq 1$, the following holds*

$$
\|\nabla f(\boldsymbol{\theta}_{t-1})\| \leq G. \tag{65}
$$

*Proof.*

$$
\|\nabla f(\boldsymbol{\theta}_{t-1})\| = \sqrt{\|\mathbb{E}[\boldsymbol{g}_t]\|^2} \tag{66}
$$

$$
\leq \sqrt{\mathbb{E}\left[ \|\boldsymbol{g}_t\|^2 \right]} \tag{67}
$$

$$
\leq G. \tag{68}
$$

The first inequality holds because $\mathbb{E}[(\boldsymbol{g}_t)_i]^2 \leq \mathbb{E}[(\boldsymbol{g}_t)_i^2]$, and the second inequality holds due to Assumption 4. $\square$

**Lemma 5.** *For all $\boldsymbol{\theta} \in \mathbb{R}^D$ and $t \geq 1$, the following holds*

$$
\mathbb{E}\left[ \|\boldsymbol{g}_t\| \right] \leq G \tag{69}
$$

*Proof.*

$$
\mathbb{E}\left[ \|\boldsymbol{g}_t\| \right] \leq \mathbb{E}\left[ \|\boldsymbol{g}_t\|^2 \right]^{1/2} \tag{70}
$$

$$
\leq G, \tag{71}
$$

where the first inequality holds due to the Hölder's inequality and the second one holds due to Assumption 4. $\square$

**Lemma 6.** *For the RMSprop algorithm, the following holds for $t \geq 1$:*

$$
\mathbb{E}\left[ \sum_{i=1}^D (\boldsymbol{v}_t)_i \right] \leq \left(1-\beta_2^t\right) G^2 \tag{72}
$$

*Proof.*

$$\mathbb{E}\left[\sum_{i=1}^{D} (\boldsymbol{v}_t)_i\right] = \mathbb{E}\left[(1-\beta_2)\sum_{i=1}^{D}\sum_{k=1}^{t}\beta_2^{t-k}(\boldsymbol{g}_k)_i^2\right] \tag{73}$$

$$\leq (1-\beta_2)\,G^2\sum_{k=1}^{t}\beta_2^{t-k} \tag{74}$$

$$= \left(1-\beta_2^t\right)G^2. \tag{75}$$

$\square$

**Lemma 7.** *For the RMSprop algorithm, the following holds:*

$$\mathbb{E}\left[\nabla f\left(\boldsymbol{\theta}_{t-1}\right)^{\top}\left(\frac{\boldsymbol{g}_t}{\sqrt{\boldsymbol{v}_t+\epsilon^2}}\right)\right]$$
$$\geq \frac{1}{2}\mathbb{E}\left[\nabla f\left(\boldsymbol{\theta}_{t-1}\right)^{\top}\left(\frac{\boldsymbol{g}_t}{\sqrt{\tilde{\boldsymbol{v}}_t+\epsilon^2}}\right)\right] - 2G\sqrt{1-\beta_2}\mathbb{E}\left[\left\|\frac{\boldsymbol{g}_t}{\sqrt{\boldsymbol{v}_t+\epsilon^2}}\right\|^2\right] \tag{76}$$

*Proof.*

$$\mathbb{E}\left[\nabla f\left(\boldsymbol{\theta}_{t-1}\right)^{\top}\left(\frac{\boldsymbol{g}_t}{\sqrt{\boldsymbol{v}_t+\epsilon^2}}\right)\right] = \sum_{i=1}^{D}\mathbb{E}\left[\frac{\left(\nabla f\left(\boldsymbol{\theta}_{t-1}\right)\right)_i(\boldsymbol{g}_t)_i}{\sqrt{(\boldsymbol{v}_t)_i+\epsilon^2}}\right] \tag{77}$$

We define $\tilde{\boldsymbol{v}}_t$ as follows:

$$\tilde{\boldsymbol{v}}_t = \beta_2\boldsymbol{v}_{t-1} + (1-\beta_2)\,\mathbb{E}\left[\boldsymbol{g}_t\odot\boldsymbol{g}_t\right] \tag{78}$$

Using this, the following holds:

$$\mathbb{E}\left[\frac{\left(\nabla f\left(\boldsymbol{\theta}_{t-1}\right)\right)_i(\boldsymbol{g}_t)_i}{\sqrt{(\boldsymbol{v}_t)_i+\epsilon^2}}\right]$$
$$= \mathbb{E}\left[\frac{\left(\nabla f\left(\boldsymbol{\theta}_{t-1}\right)\right)_i(\boldsymbol{g}_t)_i}{\sqrt{(\tilde{\boldsymbol{v}}_t)_i+\epsilon^2}}\right] + \mathbb{E}\left[\left(\nabla f\left(\boldsymbol{\theta}_{t-1}\right)\right)_i(\boldsymbol{g}_t)_i\left(\frac{1}{\sqrt{(\boldsymbol{v}_t)_i+\epsilon^2}}-\frac{1}{\sqrt{(\tilde{\boldsymbol{v}}_t)_i+\epsilon^2}}\right)\right] \tag{79}$$

$$= \mathbb{E}\left[\frac{\left(\nabla f\left(\boldsymbol{\theta}_{t-1}\right)\right)_i^2}{\sqrt{(\tilde{\boldsymbol{v}}_t)_i+\epsilon^2}}\right] + \mathbb{E}\left[\left(\nabla f\left(\boldsymbol{\theta}_{t-1}\right)\right)_i(\boldsymbol{g}_t)_i\left(\frac{1}{\sqrt{(\boldsymbol{v}_t)_i+\epsilon^2}}-\frac{1}{\sqrt{(\tilde{\boldsymbol{v}}_t)_i+\epsilon^2}}\right)\right] \tag{80}$$

$$\geq \mathbb{E}\left[\frac{\left(\nabla f\left(\boldsymbol{\theta}_{t-1}\right)\right)_i^2}{\sqrt{(\tilde{\boldsymbol{v}}_t)_i+\epsilon^2}}\right] - \mathbb{E}\left[\left|\left(\nabla f\left(\boldsymbol{\theta}_{t-1}\right)\right)_i(\boldsymbol{g}_t)_i\left(\frac{1}{\sqrt{(\boldsymbol{v}_t)_i+\epsilon^2}}-\frac{1}{\sqrt{(\tilde{\boldsymbol{v}}_t)_i+\epsilon^2}}\right)\right|\right], \tag{81}$$

where the last inequality holds due to $A \geq -|A|$. For the second term, the following holds:

$$\left|\left(\nabla f\left(\boldsymbol{\theta}_{t-1}\right)\right)_i(\boldsymbol{g}_t)_i\left(\frac{1}{\sqrt{(\boldsymbol{v}_t)_i+\epsilon^2}}-\frac{1}{\sqrt{(\tilde{\boldsymbol{v}}_t)_i+\epsilon^2}}\right)\right|$$
$$= (1-\beta_2)\left|\left(\nabla f\left(\boldsymbol{\theta}_{t-1}\right)\right)_i(\boldsymbol{g}_t)_i\frac{\mathbb{E}\left[(\boldsymbol{g}_t)_i^2\right]-(\boldsymbol{g}_t)_i^2}{\sqrt{(\boldsymbol{v}_t)_i+\epsilon^2}\sqrt{(\tilde{\boldsymbol{v}}_t)_i+\epsilon^2}\left(\sqrt{(\boldsymbol{v}_t)_i+\epsilon^2}+\sqrt{(\tilde{\boldsymbol{v}}_t)_i+\epsilon^2}\right)}\right| \tag{82}$$

$$\leq (1-\beta_2)\left(\frac{\left|\left(\nabla f\left(\boldsymbol{\theta}_{t-1}\right)\right)_i(\boldsymbol{g}_t)_i\right|\mathbb{E}\left[(\boldsymbol{g}_t)_i^2\right]}{\sqrt{(\boldsymbol{v}_t)_i+\epsilon^2}\left((\tilde{\boldsymbol{v}}_t)_i+\epsilon^2\right)}+\frac{\left|\left(\nabla f\left(\boldsymbol{\theta}_{t-1}\right)\right)_i(\boldsymbol{g}_t)_i\right|(\boldsymbol{g}_t)_i^2}{\left((\boldsymbol{v}_t)_i+\epsilon^2\right)\sqrt{(\tilde{\boldsymbol{v}}_t)_i+\epsilon^2}}\right), \tag{83}$$

where the last inequality holds due to the triangle inequality. For the first term, the following holds:

$$\mathbb{E}\left[\frac{|(\nabla f(\boldsymbol{\theta}_{t-1}))_i(\boldsymbol{g}_t)_i|\mathbb{E}\left[(\boldsymbol{g}_t)_i^2\right]}{\sqrt{(\boldsymbol{v}_t)_i+\epsilon^2}((\tilde{\boldsymbol{v}}_t)_i+\epsilon^2)}\right]$$

$$\leq \frac{1}{(1-\beta_2)}\mathbb{E}\left[\frac{(\nabla f(\boldsymbol{\theta}_{t-1}))_i^2}{4\sqrt{(\tilde{\boldsymbol{v}}_t)_i+\epsilon^2}}\right]+(1-\beta_2)\mathbb{E}\left[\frac{(\boldsymbol{g}_t)_i^2\mathbb{E}\left[(\boldsymbol{g}_t)_i^2\right]^2}{((\boldsymbol{v}_t)_i+\epsilon^2)((\tilde{\boldsymbol{v}}_t)_i+\epsilon^2)^{3/2}}\right] \quad (84)$$

$$\leq \frac{1}{(1-\beta_2)}\mathbb{E}\left[\frac{(\nabla f(\boldsymbol{\theta}_{t-1}))_i^2}{4\sqrt{(\tilde{\boldsymbol{v}}_t)_i+\epsilon^2}}\right]+\mathbb{E}\left[\frac{(\boldsymbol{g}_t)_i^2\sqrt{\mathbb{E}\left[(\boldsymbol{g}_t)_i^2\right]}}{\sqrt{1-\beta_2}((\boldsymbol{v}_t)_i+\epsilon^2)}\right] \quad (85)$$

$$\leq \frac{1}{(1-\beta_2)}\mathbb{E}\left[\frac{(\nabla f(\boldsymbol{\theta}_{t-1}))_i^2}{4\sqrt{(\tilde{\boldsymbol{v}}_t)_i+\epsilon^2}}\right]+\frac{G}{\sqrt{1-\beta_2}}\mathbb{E}\left[\frac{(\boldsymbol{g}_t)_i^2}{(\boldsymbol{v}_t)_i+\epsilon^2}\right] \quad (86)$$

The first inequality is derived using the following fact:

$$\forall \lambda > 0, x, y \in \mathbb{R}, xy \leq \frac{\lambda}{2}x^2+\frac{y^2}{2\lambda}. \quad (87)$$

For the second term of Eq. (83), the following holds:

$$\mathbb{E}\left[\frac{|(\nabla f(\boldsymbol{\theta}_{t-1}))_i(\boldsymbol{g}_t)_i|(\boldsymbol{g}_t)_i^2}{((\boldsymbol{v}_t)_i+\epsilon^2)\sqrt{(\tilde{\boldsymbol{v}}_t)_i+\epsilon^2}}\right] \quad (88)$$

$$\leq \frac{1}{(1-\beta_2)}\mathbb{E}\left[\frac{(\nabla f(\boldsymbol{\theta}_{t-1}))_i^2}{4\sqrt{(\tilde{\boldsymbol{v}}_t)_i+\epsilon^2}}\frac{(\boldsymbol{g}_t)_i^2}{\mathbb{E}\left[(\boldsymbol{g}_t)_i^2\right]}\right]+(1-\beta_2)\mathbb{E}\left[\frac{\mathbb{E}\left[(\boldsymbol{g}_t)_i^2\right]}{\sqrt{(\tilde{\boldsymbol{v}}_t)_i+\epsilon^2}}\frac{(\boldsymbol{g}_t)_i^4}{((\tilde{\boldsymbol{v}}_t)_i+\epsilon^2)^2}\right] \quad (89)$$

$$\leq \frac{1}{(1-\beta_2)}\mathbb{E}\left[\frac{(\nabla f(\boldsymbol{\theta}_{t-1}))_i^2}{4\sqrt{(\tilde{\boldsymbol{v}}_t)_i+\epsilon^2}}\right]+\mathbb{E}\left[\frac{\sqrt{\mathbb{E}\left[(\boldsymbol{g}_t)_i^2\right]}(\boldsymbol{g}_t)_i^2}{\sqrt{1-\beta_2}((\tilde{\boldsymbol{v}}_t)_i+\epsilon^2)}\right] \quad (90)$$

$$\leq \frac{1}{(1-\beta_2)}\mathbb{E}\left[\frac{(\nabla f(\boldsymbol{\theta}_{t-1}))_i^2}{4\sqrt{(\tilde{\boldsymbol{v}}_t)_i+\epsilon^2}}\right]+\frac{G}{\sqrt{1-\beta_2}}\mathbb{E}\left[\frac{(\boldsymbol{g}_t)_i^2}{(\boldsymbol{v}_t)_i+\epsilon^2}\right] \quad (91)$$

The first inequality is derived using Eq. (87).

Putting these inequalities together, the following is derived:

$$\mathbb{E}\left[\nabla f(\boldsymbol{\theta}_{t-1})^\top\left(\frac{\boldsymbol{g}_t}{\sqrt{\boldsymbol{v}_t+\epsilon^2}}\right)\right]$$

$$\geq \sum_{i=1}^{D}\mathbb{E}\left[\frac{(\nabla f(\boldsymbol{\theta}_{t-1}))_i^2}{2\sqrt{(\tilde{\boldsymbol{v}}_t)_i+\epsilon^2}}\right]-2G\sqrt{1-\beta_2}\mathbb{E}\left[\frac{(\boldsymbol{g}_t)_i^2}{(\boldsymbol{v}_t)_i+\epsilon^2}\right] \quad (92)$$

$$\geq \frac{1}{2}\mathbb{E}\left[\nabla f(\boldsymbol{\theta}_{t-1})^\top\left(\frac{\boldsymbol{g}_t}{\sqrt{\tilde{\boldsymbol{v}}_t+\epsilon^2}}\right)\right]-2G\sqrt{1-\beta_2}\mathbb{E}\left[\left\|\frac{\boldsymbol{g}_t}{\sqrt{\boldsymbol{v}_t+\epsilon^2}}\right\|^2\right]. \quad (93)$$

$\square$

**Lemma 8.** *For the RMSprop algorithm, the following holds:*

$$\sum_{t=1}^{T}\mathbb{E}\left[\left\|\frac{\boldsymbol{g}_t}{\sqrt{\boldsymbol{v}_t+\epsilon^2}}\right\|^2\right]\leq D\left(\log\left(1+\frac{(1-\beta_2^T)G^2}{\epsilon^2}\right)-T\log\beta_2\right) \quad (94)$$

*Proof.*

$$\left\| \frac{\boldsymbol{g}_t}{\sqrt{\boldsymbol{v}_t + \epsilon^2}} \right\|^2 = \sum_{i=1}^{D} \frac{(\boldsymbol{g}_t)_i^2}{(\boldsymbol{v}_t)_i + \epsilon^2} \tag{95}$$

$$\frac{(\boldsymbol{g}_t)_i^2}{(\boldsymbol{v}_t)_i + \epsilon^2} = \frac{1}{1 - \beta_2} \frac{(1 - \beta_2)(\boldsymbol{g}_t)_i^2}{(\boldsymbol{v}_t)_i + \epsilon^2} \tag{96}$$

$$\leq -\frac{1}{1 - \beta_2} \log \left( 1 - \frac{(1 - \beta_2)(\boldsymbol{g}_t)_i^2}{(\boldsymbol{v}_t)_i + \epsilon^2} \right) \tag{97}$$

$$= \frac{1}{1 - \beta_2} \log \left( \frac{(\boldsymbol{v}_t)_i + \epsilon^2}{\beta_2 (\boldsymbol{v}_{t-1})_i + \epsilon^2} \right) \tag{98}$$

$$= \frac{1}{1 - \beta_2} \left( \log \left( \frac{(\boldsymbol{v}_t)_i + \epsilon^2}{(\boldsymbol{v}_{t-1})_i + \epsilon^2} \right) + \log \left( \frac{(\boldsymbol{v}_{t-1})_i + \epsilon^2}{\beta_2 (\boldsymbol{v}_{t-1})_i + \epsilon^2} \right) \right) \tag{99}$$

$$\leq \frac{1}{1 - \beta_2} \left( \log \left( \frac{(\boldsymbol{v}_t)_i + \epsilon^2}{(\boldsymbol{v}_{t-1})_i + \epsilon^2} \right) - \log \beta_2 \right) \tag{100}$$

$$\sum_{t=1}^{T} \frac{(\boldsymbol{g}_t)_i^2}{(\boldsymbol{v}_t)_i + \epsilon^2} \leq \frac{1}{1 - \beta_2} \left( \log \left( \frac{(\boldsymbol{v}_T)_i + \epsilon^2}{\epsilon^2} \right) - T \log \beta_2 \right) \tag{101}$$

$$\leq \frac{1}{1 - \beta_2} \left( \log \left( 1 + \frac{(1 - \beta_2^T) G^2}{\epsilon^2} \right) - T \log \beta_2 \right) \tag{102}$$

$$\sum_{t=1}^{T} \mathbb{E} \left[ \left\| \frac{\boldsymbol{g}_t}{\sqrt{\boldsymbol{v}_t + \epsilon^2}} \right\|^2 \right] \leq \sum_{i=1}^{D} \mathbb{E} \left[ \sum_{t=1}^{T} \frac{(\boldsymbol{g}_t)_i^2}{(\boldsymbol{v}_t)_i + \epsilon^2} \right] \tag{103}$$

$$\leq \frac{1}{1 - \beta_2} \sum_{i=1}^{D} \mathbb{E} \left[ \log \left( 1 + \frac{(\boldsymbol{v}_T)_i}{\epsilon^2} \right) \right] - \frac{DT \log \beta_2}{1 - \beta_2} \tag{104}$$

$$\leq \sum_{i=1}^{D} \log \left( 1 + \frac{\mathbb{E}\left[ (\boldsymbol{v}_T)_i \right]}{\epsilon^2} \right) - \frac{DT \log \beta_2}{1 - \beta_2} \tag{105}$$

$$\leq \frac{D}{1 - \beta_2} \left( \log \left( 1 + \frac{(1 - \beta_2^T) G^2}{\epsilon^2} \right) - T \log \beta_2 \right) \tag{106}$$

$\square$

**Lemma 9.** *For the RMSprop algorithm, the following holds:*

$$\mathbb{E} \left[ \nabla f(\boldsymbol{\theta}_{t-1})^\top \left( \frac{\boldsymbol{g}_t}{\sqrt{\beta_2 \tilde{\boldsymbol{v}}_t + \epsilon^2}} \right) \right] \geq \frac{\mathbb{E} \left[ \|\nabla f(\boldsymbol{\theta}_{t-1})\|^{4/3} \right]^{3/2}}{\sqrt{(1 - \beta_2^t) G^2 + \epsilon^2}} \tag{107}$$

*Proof.*

$$\mathbb{E}\left[\nabla f\left(\boldsymbol{\theta}_{t-1}\right)^{\top}\left(\frac{\boldsymbol{g}_t}{\sqrt{\tilde{\boldsymbol{v}}_t+\epsilon^2}}\right)\right]$$

$$=\sum_{i=1}^{D}\mathbb{E}\left[\frac{\left(\nabla f\left(\boldsymbol{\theta}_{t-1}\right)\right)_i\cdot\left(\boldsymbol{g}_t\right)_i}{\sqrt{\left(\tilde{\boldsymbol{v}}_t\right)_i+\epsilon^2}}\right]$$

$$=\sum_{i=1}^{D}\mathbb{E}\left[\frac{\left(\nabla f\left(\boldsymbol{\theta}_{t-1}\right)\right)_i^2}{\sqrt{\beta_2\left(\boldsymbol{v}_{t-1}\right)_i+\epsilon^2}}\right]$$

$$\geq\mathbb{E}\left[\frac{\left\|\nabla f\left(\boldsymbol{\theta}_{t-1}\right)\right\|^2}{\sqrt{\sum_{i=1}^{D}\left(\tilde{\boldsymbol{v}}_t\right)_i+\epsilon^2}}\right]$$

$$\geq\frac{\mathbb{E}\left[\left\|\nabla f\left(\boldsymbol{\theta}_{t-1}\right)\right\|^{4/3}\right]^{3/2}}{\sqrt{\mathbb{E}\left[\sum_{i=1}^{D}\left(\tilde{\boldsymbol{v}}_t\right)_i\right]+\epsilon^2}}$$

$$\geq\frac{\mathbb{E}\left[\left\|\nabla f\left(\boldsymbol{\theta}_{t-1}\right)\right\|^{4/3}\right]^{3/2}}{\sqrt{\left(1-\beta_2^t\right)G^2+\epsilon^2}}. \tag{108}$$

The second equality holds due to Assumption 2. The first inequality holds because $\left(\tilde{\boldsymbol{v}}_t\right)_i\geq 0$ for all $i=1,\ldots,D$. The second inequality holds due to the Hölder's inequality. The last inequality holds due to Lemma 6. $\qquad\square$

**Lemma 10.** *For the ADOPT algorithm, the following holds for $t\geq 1$:*

$$\boldsymbol{\phi}_t-\boldsymbol{\phi}_{t-1}=\frac{\left(\alpha_{t-1}-\alpha_t\right)\beta_1}{1-\beta_1}\boldsymbol{m}_{t-1}-\alpha_t\frac{\boldsymbol{g}_t}{\sqrt{\boldsymbol{v}_{t-1}+\epsilon^2}}, \tag{109}$$

*where we define $\alpha_0=\alpha$.*

*Proof.* For $t=1$, the following holds by definition:

$$\boldsymbol{\phi}_1-\boldsymbol{\phi}_0=\frac{1}{1-\beta_1}\boldsymbol{\theta}_1-\left(\frac{\beta_1}{1-\beta_1}+1\right)\boldsymbol{\theta}_0 \tag{110}$$

$$=\frac{1}{1-\beta_1}\left(\boldsymbol{\theta}_1-\boldsymbol{\theta}_0\right) \tag{111}$$

$$=-\frac{\alpha_1\boldsymbol{g}_1}{\sqrt{\boldsymbol{v}_0+\epsilon^2}}. \tag{112}$$

For $t\geq 2$, the following holds:

$$\boldsymbol{\phi}_t-\boldsymbol{\phi}_{t-1}=\frac{1}{1-\beta_1}\left(\boldsymbol{\theta}_t-\boldsymbol{\theta}_{t-1}\right)-\frac{\beta_1}{1-\beta_1}\left(\boldsymbol{\theta}_{t-1}-\boldsymbol{\theta}_{t-2}\right) \tag{113}$$

$$=\frac{1}{1-\beta_1}\left(\alpha_{t-1}\beta_1\boldsymbol{m}_{t-1}-\alpha_t\boldsymbol{m}_t\right) \tag{114}$$

$$=\frac{1}{1-\beta_1}\left(\alpha_{t-1}\beta_1\boldsymbol{m}_{t-1}-\alpha_t\left(\beta_1\boldsymbol{m}_{t-1}+\left(1-\beta_1\right)\frac{\boldsymbol{g}_t}{\sqrt{\boldsymbol{v}_{t-1}+\epsilon^2}}\right)\right) \tag{115}$$

$$=\frac{1}{1-\beta_1}\left(\left(\alpha_{t-1}-\alpha_t\right)\beta_1\boldsymbol{m}_{t-1}-\alpha_t\left(1-\beta_1\right)\frac{\boldsymbol{g}_t}{\sqrt{\boldsymbol{v}_{t-1}+\epsilon^2}}\right) \tag{116}$$

$$=\frac{\left(\alpha_{t-1}-\alpha_t\right)\beta_1}{1-\beta_1}\boldsymbol{m}_{t-1}-\alpha_t\frac{\boldsymbol{g}_t}{\sqrt{\boldsymbol{v}_{t-1}+\epsilon^2}} \tag{117}$$

$$\square$$

**Lemma 11.** *For the ADOPT algorithm, the following holds for $t \geq 1$:*

$$\phi_{t-1} - \theta_{t-1} = -\frac{\alpha_{t-1}\beta_1}{1-\beta_1} m_{t-1}. \tag{118}$$

*Proof.* For $t = 1$, Eq. (118) holds obviously because $\phi_0 = \theta_0$ and $m_0 = 0$. For $t \geq 2$, the following holds:

$$\phi_{t-1} - \theta_{t-1} = \left(\frac{1}{1-\beta_1} - 1\right)\theta_{t-1} - \frac{\beta_1}{1-\beta_1}\theta_{t-2} \tag{119}$$

$$= \frac{\beta_1}{1-\beta_1}(\theta_{t-1} - \theta_{t-2}) \tag{120}$$

$$= -\frac{\alpha_{t-1}\beta_1}{1-\beta_1} m_{t-1}. \tag{121}$$

$\square$

**Lemma 12.** *For the ADOPT algorithm, the following holds for $t \geq 1$:*

$$\mathbb{E}\left[\nabla f(\theta_{t-1})^\top (\phi_t - \phi_{t-1})\right]$$
$$\leq \frac{(\alpha_{t-1} - \alpha_t)\beta_1\left(1-\beta_1^{t-1}\right)G^2}{(1-\beta_1)\sqrt{\beta_2^{t-2}+\epsilon^2}} - \alpha_t \frac{\mathbb{E}\left[\|\nabla f(\theta_{t-1})\|_i^{4/3}\right]^{3/2}}{\sqrt{\max\left\{G^2 + (1-G^2)\beta_2^T, 1\right\} + \epsilon^2}}. \tag{122}$$

*Proof.*

$$\nabla f(\theta_{t-1})^\top (\phi_t - \phi_{t-1})$$
$$= \frac{(\alpha_{t-1} - \alpha_t)\beta_1}{1-\beta_1}\nabla f(\theta_{t-1}))^\top m_{t-1} - \alpha_t \nabla f(\theta_{t-1})^\top \frac{g_t}{\sqrt{v_{t-1}+\epsilon^2}} \tag{123}$$

$$\leq \frac{(\alpha_{t-1} - \alpha_t)\beta_1}{1-\beta_1}\|\nabla f(\theta_{t-1}))\|\|m_{t-1}\| - \alpha_t \nabla f(\theta_{t-1})^\top \frac{g_t}{\sqrt{v_{t-1}+\epsilon^2}} \tag{124}$$

$$\leq \frac{(\alpha_{t-1} - \alpha_t)\beta_1 G}{1-\beta_1}\|m_{t-1}\| - \alpha_t \nabla f(\theta_{t-1})^\top \frac{g_t}{\sqrt{v_{t-1}+\epsilon^2}}. \tag{125}$$

By taking the expectation for both sides, the following holds:

$$
\mathbb{E}\left[\nabla f\left(\boldsymbol{\theta}_{t-1}\right)^{\top}\cdot\left(\boldsymbol{\phi}_{t}-\boldsymbol{\phi}_{t-1}\right)\right]
$$

$$
\leq \frac{\left(\alpha_{t-1}-\alpha_{t}\right)\beta_{1}G}{1-\beta_{1}}\mathbb{E}\left[\|\boldsymbol{m}_{t-1}\|\right]-\alpha_{t}\mathbb{E}\left[\nabla f\left(\boldsymbol{\theta}_{t-1}\right)^{\top}\frac{\boldsymbol{g}_{t}}{\sqrt{\boldsymbol{v}_{t-1}+\epsilon^{2}}}\right] \tag{126}
$$

$$
\leq \frac{\left(\alpha_{t-1}-\alpha_{t}\right)\beta_{1}G}{1-\beta_{1}}\mathbb{E}\left[\|\boldsymbol{m}_{t-1}\|\right]-\alpha_{t}\sum_{i=1}^{D}\mathbb{E}\left[\frac{\left(\nabla f\left(\boldsymbol{\theta}_{t-1}\right)\right)_{i}\cdot\left(\boldsymbol{g}_{t}\right)_{i}}{\sqrt{\left(\boldsymbol{v}_{t-1}\right)_{i}+\epsilon^{2}}}\right] \tag{127}
$$

$$
\leq \frac{\left(\alpha_{t-1}-\alpha_{t}\right)\beta_{1}G}{1-\beta_{1}}\mathbb{E}\left[\|\boldsymbol{m}_{t-1}\|\right]-\alpha_{t}\sum_{i=1}^{D}\mathbb{E}\left[\frac{\left(\nabla f\left(\boldsymbol{\theta}_{t-1}\right)\right)_{i}^{2}}{\sqrt{\left(\boldsymbol{v}_{t-1}\right)_{i}+\epsilon^{2}}}\right] \tag{128}
$$

$$
\leq \frac{\left(\alpha_{t-1}-\alpha_{t}\right)\beta_{1}G}{1-\beta_{1}}\mathbb{E}\left[\|\boldsymbol{m}_{t-1}\|\right]-\alpha_{t}\mathbb{E}\left[\frac{\|\nabla f\left(\boldsymbol{\theta}_{t-1}\right)\|^{2}}{\sqrt{\sum_{i=1}^{D}\left(\boldsymbol{v}_{t-1}\right)_{i}+\epsilon^{2}}}\right] \tag{129}
$$

$$
\leq \frac{\left(\alpha_{t-1}-\alpha_{t}\right)\beta_{1}G}{1-\beta_{1}}\mathbb{E}\left[\|\boldsymbol{m}_{t-1}\|\right]-\alpha_{t}\frac{\mathbb{E}\left[\|\nabla f\left(\boldsymbol{\theta}_{t-1}\right)\|_{i}^{4/3}\right]^{3/2}}{\sqrt{\mathbb{E}\left[\sum_{i=1}^{D}\left(\boldsymbol{v}_{t-1}\right)_{i}\right]+\epsilon^{2}}} \tag{130}
$$

$$
\leq \frac{\left(\alpha_{t-1}-\alpha_{t}\right)\beta_{1}G}{1-\beta_{1}}\mathbb{E}\left[\|\boldsymbol{m}_{t-1}\|\right]-\alpha_{t}\frac{\mathbb{E}\left[\|\nabla f\left(\boldsymbol{\theta}_{t-1}\right)\|_{i}^{4/3}\right]^{3/2}}{\sqrt{\max\left\{G^{2}+(1-G^{2})\beta_{2}^{T},1\right\}+\epsilon^{2}}} \tag{131}
$$

$$
\leq \frac{\left(\alpha_{t-1}-\alpha_{t}\right)\beta_{1}\left(1-\beta_{1}^{t-1}\right)G^{2}}{(1-\beta_{1})\sqrt{\beta_{2}^{t-2}+\epsilon^{2}}}-\alpha_{t}\frac{\mathbb{E}\left[\|\nabla f\left(\boldsymbol{\theta}_{t-1}\right)\|_{i}^{4/3}\right]^{3/2}}{\sqrt{\max\left\{G^{2}+(1-G^{2})\beta_{2}^{T},1\right\}+\epsilon^{2}}}. \tag{132}
$$

$\square$

**Lemma 13.** *For the ADOPT algorithm, the following holds for $t \geq 0$:*

$$
\mathbb{E}\left[\sum_{i=1}^{D}\left(\boldsymbol{v}_{t}\right)_{i}\right] \leq \max\left\{G^{2}+(1-G^{2})\beta_{2}^{T},1\right\}. \tag{133}
$$

*Proof.*

$$
\mathbb{E}\left[\sum_{i=1}^{D}\left(\boldsymbol{v}_{t}\right)_{i}\right] = \mathbb{E}\left[\beta_{2}^{t}+(1-\beta_{2})\sum_{i=1}^{D}\sum_{k=1}^{t}\beta_{2}^{t-k}\left(\boldsymbol{g}_{k-1}\right)_{i}^{2}\right] \tag{134}
$$

$$
\leq \beta_{2}^{t}+(1-\beta_{2})G^{2}\sum_{k=1}^{t}\beta_{2}^{t-k} \tag{135}
$$

$$
= \beta_{2}^{t}+\left(1-\beta_{2}^{t}\right)G^{2} \tag{136}
$$

$$
= G^{2}+(1-G^{2})\beta_{2}^{t}. \tag{137}
$$

When $G < 1$, the following holds:

$$
\mathbb{E}\left[\sum_{i=1}^{D}\left(\boldsymbol{v}_{t}\right)_{i}\right] \leq G^{2}+(1-G^{2})\beta_{2}^{t} \tag{138}
$$

$$
\leq 1. \tag{139}
$$

When $G \geq 1$, the following holds:

$$\mathbb{E}\left[\sum_{i=1}^{D} (\boldsymbol{v}_t)_i\right] \leq G^2 + (1 - G^2)\beta_2^t \tag{140}$$

$$\leq G^2 + (1 - G^2)\beta_2^T \tag{141}$$

$$\leq G^2. \tag{142}$$

Putting the two together yields:

$$\mathbb{E}\left[\sum_{i=1}^{D} (\boldsymbol{v}_t)_i\right] \leq \max\left\{G^2 + (1 - G^2)\beta_2^T, 1\right\} \tag{143}$$

$$\leq \max\left\{G^2, 1\right\}. \tag{144}$$

$\square$

**Lemma 14.** *For the ADOPT algorithm, the following holds for $0 \leq t \leq T$.*

$$\mathbb{E}\left[\|\boldsymbol{m}_t\|^2\right] \leq \frac{G^2}{\epsilon^2}. \tag{145}$$

*Proof.*

$$\mathbb{E}\left[\|\boldsymbol{m}_t\|^2\right] = \mathbb{E}\left[\left\|\beta_1 \boldsymbol{m}_{t-1} + (1 - \beta_1)\frac{\boldsymbol{g}_t}{\sqrt{\boldsymbol{v}_{t-1} + \epsilon^2}}\right\|^2\right] \tag{146}$$

$$= \mathbb{E}\left[\beta_1^2 \|\boldsymbol{m}_{t-1}\|^2 + (1 - \beta_1)^2 \left\|\frac{\boldsymbol{g}_t}{\sqrt{\boldsymbol{v}_{t-1} + \epsilon^2}}\right\|^2 + 2\beta_1(1 - \beta_1)\boldsymbol{m}_{t-1}^\top \frac{\boldsymbol{g}_t}{\sqrt{\boldsymbol{v}_{t-1} + \epsilon^2}}\right] \tag{147}$$

$$\leq \mathbb{E}\left[\beta_1 \|\boldsymbol{m}_{t-1}\|^2 + (1 - \beta_1)\left\|\frac{\boldsymbol{g}_t}{\sqrt{\boldsymbol{v}_{t-1} + \epsilon^2}}\right\|^2\right] \tag{148}$$

$$\leq \mathbb{E}\left[\beta_1 \|\boldsymbol{m}_{t-1}\|^2 + \frac{1 - \beta_1}{\beta_2^{t-1} + \epsilon^2}\|\boldsymbol{g}_t\|^2\right] \tag{149}$$

$$\leq \mathbb{E}\left[\frac{1 - \beta_1}{\beta_2^{t-1} + \epsilon^2}\sum_{k=1}^{t}\beta_1^{t-k}\|\boldsymbol{g}_k\|^2\right] \tag{150}$$

$$\leq \frac{(1 - \beta_1)G^2}{\beta_2^{t-1} + \epsilon^2}\sum_{k=1}^{t}\beta_1^{t-k} \tag{151}$$

$$\leq \frac{(1 - \beta_1^t)G^2}{\beta_2^{t-1} + \epsilon^2} \tag{152}$$

$$\leq \frac{G^2}{\epsilon^2}. \tag{153}$$

$\square$

**Lemma 15.** *For the ADOPT algorithm, the following holds for $t \geq 0$.*

$$\mathbb{E}\left[\|\boldsymbol{m}_t\|\right] \leq \frac{G}{\epsilon} \tag{154}$$

*Proof.*

$$\mathbb{E}\left[\|\boldsymbol{m}_t\|\right] = \mathbb{E}\left[\left\|(1-\beta_1)\sum_{k=1}^t \beta_1^{t-k}\frac{\boldsymbol{g}_k}{\sqrt{\boldsymbol{v}_{k-1}+\epsilon^2}}\right\|\right] \tag{155}$$

$$\leq (1-\beta_1)\sum_{k=1}^t \beta_1^{t-k}\mathbb{E}\left[\left\|\frac{\boldsymbol{g}_k}{\sqrt{\boldsymbol{v}_{k-1}+\epsilon^2}}\right\|\right] \tag{156}$$

$$\leq (1-\beta_1)\sum_{k=1}^t \frac{\beta_1^{t-k}}{\sqrt{\beta_2^{k-1}+\epsilon^2}}\mathbb{E}\left[\|\boldsymbol{g}_k\|\right] \tag{157}$$

$$\leq \frac{1-\beta_1}{\sqrt{\beta_2^{t-1}+\epsilon^2}}\sum_{k=1}^t \beta_1^{t-k}\mathbb{E}\left[\|\boldsymbol{g}_k\|^2\right]^{1/2} \tag{158}$$

$$\leq \frac{(1-\beta_1)\,G}{\sqrt{\beta_2^{t-1}+\epsilon^2}}\sum_{k=1}^t \beta_1^{t-k} \tag{159}$$

$$= \frac{(1-\beta_1^t)\,G}{\sqrt{\beta_2^{t-1}+\epsilon^2}} \tag{160}$$

$$\leq \frac{G}{\epsilon}. \tag{161}$$

$\square$

**Lemma 16.** *For the ADOPT algorithm, the following holds for $t \geq 1$:*

$$\mathbb{E}\left[\|\boldsymbol{\phi}_{t-1}-\boldsymbol{\theta}_{t-1}\|\,\|\boldsymbol{\phi}_t-\boldsymbol{\phi}_{t-1}\|\right]$$

$$\leq \frac{\alpha_{t-1}\left(\alpha_{t-1}-\alpha_t\right)\beta_1^2\left(1-\beta_1^{t-1}\right)G^2}{\left(\beta_2^{t-2}+\epsilon^2\right)\left(1-\beta_1\right)^2} + \frac{\alpha_t\alpha_{t-1}\beta_1\sqrt{1-\beta_1^{t-1}}G^2}{\left(1-\beta_1\right)\sqrt{\beta_2^{t-1}+\epsilon^2}\sqrt{\beta_2^{t-2}+\epsilon^2}}. \tag{162}$$

*Proof.*

$$\|\boldsymbol{\phi}_{t-1}-\boldsymbol{\theta}_{t-1}\|\,\|\boldsymbol{\phi}_t-\boldsymbol{\phi}_{t-1}\|$$

$$= \left\|-\frac{\alpha_{t-1}\beta_1}{1-\beta_1}\boldsymbol{m}_{t-1}\right\|\left\|\frac{(\alpha_{t-1}-\alpha_t)\beta_1}{1-\beta_1}\boldsymbol{m}_{t-1}-\alpha_t\frac{\boldsymbol{g}_t}{\sqrt{\boldsymbol{v}_{t-1}+\epsilon^2}}\right\| \tag{163}$$

$$\leq \frac{\alpha_{t-1}\beta_1}{1-\beta_1}\|\boldsymbol{m}_{t-1}\|\left(\frac{(\alpha_{t-1}-\alpha_t)\beta_1}{1-\beta_1}\|\boldsymbol{m}_{t-1}\|+\alpha_t\left\|\frac{\boldsymbol{g}_t}{\sqrt{\boldsymbol{v}_{t-1}+\epsilon^2}}\right\|\right) \tag{164}$$

$$\leq \frac{\alpha_{t-1}\left(\alpha_{t-1}-\alpha_t\right)\beta_1^2}{\left(1-\beta_1\right)^2}\|\boldsymbol{m}_{t-1}\|^2 + \frac{\alpha_t\alpha_{t-1}\beta_1}{1-\beta_1}\|\boldsymbol{m}_{t-1}\|\left\|\frac{\boldsymbol{g}_t}{\sqrt{\boldsymbol{v}_{t-1}+\epsilon^2}}\right\|. \tag{165}$$

Taking the expectation yields:

$$\mathbb{E}\left[\|\boldsymbol{\phi}_{t-1}-\boldsymbol{\theta}_{t-1}\|\,\|\boldsymbol{\phi}_t-\boldsymbol{\phi}_{t-1}\|\right]$$

$$\leq \frac{\alpha_{t-1}\left(\alpha_{t-1}-\alpha_t\right)\beta_1^2}{\left(1-\beta_1\right)^2}\mathbb{E}\left[\|\boldsymbol{m}_{t-1}\|^2\right] + \frac{\alpha_t\alpha_{t-1}\beta_1}{1-\beta_1}\mathbb{E}\left[\|\boldsymbol{m}_{t-1}\|\left\|\frac{\boldsymbol{g}_t}{\sqrt{\boldsymbol{v}_{t-1}+\epsilon^2}}\right\|\right] \tag{166}$$

$$\leq \frac{\alpha_{t-1}\left(\alpha_{t-1}-\alpha_t\right)\beta_1^2}{\left(1-\beta_1\right)^2}\mathbb{E}\left[\|\boldsymbol{m}_{t-1}\|^2\right] + \frac{\alpha_t\alpha_{t-1}\beta_1}{\left(1-\beta_1\right)\sqrt{\beta_2^{t-1}+\epsilon^2}}\mathbb{E}\left[\|\boldsymbol{m}_{t-1}\|\,\|\boldsymbol{g}_t\|\right] \tag{167}$$

$$\leq \frac{\alpha_{t-1}\left(\alpha_{t-1}-\alpha_t\right)\beta_1^2\left(1-\beta_1^{t-1}\right)G^2}{\left(\beta_2^{t-2}+\epsilon^2\right)\left(1-\beta_1\right)^2} + \frac{\alpha_t\alpha_{t-1}\beta_1\sqrt{1-\beta_1^{t-1}}G^2}{\left(1-\beta_1\right)\sqrt{\beta_2^{t-1}+\epsilon^2}\sqrt{\beta_2^{t-2}+\epsilon^2}}. \tag{168}$$

$\square$

**Lemma 17.** *For the ADOPT algorithm, the following holds for $t \geq 1$:*

$$\mathbb{E}\left[\|\boldsymbol{\phi}_t - \boldsymbol{\phi}_{t-1}\|^2\right]$$

$$\leq \frac{(\alpha_{t-1} - \alpha_t)^2 \beta_1^2 \left(1 - \beta_1^{t-1}\right) G^2}{(1-\beta_1)^2 \left(\beta_2^{t-2} + \epsilon^2\right)} + \frac{\alpha_t^2 G^2}{\beta_2^{t-1} + \epsilon^2} + \frac{\alpha_t (\alpha_{t-1} - \alpha_t) \beta_1 \sqrt{1 - \beta_1^{t-1}} G^2}{(1-\beta_1) \sqrt{\beta_2^{t-1} + \epsilon^2} \sqrt{\beta_2^{t-2} + \epsilon^2}}. \quad (169)$$

*Proof.*

$$\|\boldsymbol{\phi}_t - \boldsymbol{\phi}_{t-1}\|^2$$

$$= \left\| \frac{(\alpha_{t-1} - \alpha_t) \beta_1}{1 - \beta_1} \boldsymbol{m}_{t-1} - \alpha_t \frac{\boldsymbol{g}_t}{\sqrt{\boldsymbol{v}_t + \epsilon^2}} \right\|^2 \quad (170)$$

$$= \frac{(\alpha_{t-1} - \alpha_t)^2 \beta_1^2}{(1-\beta_1)^2} \|\boldsymbol{m}_{t-1}\|^2 + \alpha_t^2 \left\| \frac{\boldsymbol{g}_t}{\sqrt{\boldsymbol{v}_{t-1} + \epsilon^2}} \right\|^2 - \frac{\alpha_t (\alpha_{t-1} - \alpha_t) \beta_1}{1 - \beta_1} \boldsymbol{m}_{t-1}^\top \frac{\boldsymbol{g}_t}{\sqrt{\boldsymbol{v}_{t-1} + \epsilon^2}} \quad (171)$$

$$\leq \frac{(\alpha_{t-1} - \alpha_t)^2 \beta_1^2}{(1-\beta_1)^2} \|\boldsymbol{m}_{t-1}\|^2 + \frac{\alpha_t^2}{\beta_2^{t-1} + \epsilon^2} \|\boldsymbol{g}_t\|^2 + \frac{\alpha_t (\alpha_{t-1} - \alpha_t) \beta_1}{1 - \beta_1} \|\boldsymbol{m}_{t-1}\| \left\| \frac{\boldsymbol{g}_t}{\sqrt{\boldsymbol{v}_{t-1} + \epsilon^2}} \right\| \quad (172)$$

$$\leq \frac{(\alpha_{t-1} - \alpha_t)^2 \beta_1^2}{(1-\beta_1)^2} \|\boldsymbol{m}_{t-1}\|^2 + \frac{\alpha_t^2}{\beta_2^{t-1} + \epsilon^2} \|\boldsymbol{g}_t\|^2 + \frac{\alpha_t (\alpha_{t-1} - \alpha_t) \beta_1}{(1 - \beta_1) \sqrt{\beta_2^{t-1} + \epsilon^2}} \|\boldsymbol{m}_{t-1}\| \|\boldsymbol{g}_t\|. \quad (173)$$

Taking the expectation yields:

$$\mathbb{E}\left[\|\boldsymbol{\phi}_t - \boldsymbol{\phi}_{t-1}\|^2\right]$$

$$\leq \frac{(\alpha_{t-1} - \alpha_t)^2 \beta_1^2}{(1-\beta_1)^2} \mathbb{E}\left[\|\boldsymbol{m}_{t-1}\|^2\right] + \frac{\alpha_t^2}{\beta_2^{t-1} + \epsilon^2} \mathbb{E}\left[\|\boldsymbol{g}_t\|^2\right] + \frac{\alpha_t (\alpha_{t-1} - \alpha_t) \beta_1}{(1 - \beta_1) \sqrt{\beta_2^{t-1} + \epsilon^2}} \mathbb{E}\left[\|\boldsymbol{m}_{t-1}\| \|\boldsymbol{g}_t\|\right] \quad (174)$$

$$\leq \frac{(\alpha_{t-1} - \alpha_t)^2 \beta_1^2 \left(1 - \beta_1^{t-1}\right) G^2}{(1-\beta_1)^2 \left(\beta_2^{t-2} + \epsilon^2\right)} + \frac{\alpha_t^2 G^2}{\beta_2^{t-1} + \epsilon^2} + \frac{\alpha_t (\alpha_{t-1} - \alpha_t) \beta_1}{(1 - \beta_1) \sqrt{\beta_2^{t-1} + \epsilon^2}} \mathbb{E}\left[\|\boldsymbol{m}_{t-1}\| \|\boldsymbol{g}_t\|\right] \quad (175)$$

$$\leq \frac{(\alpha_{t-1} - \alpha_t)^2 \beta_1^2 \left(1 - \beta_1^{t-1}\right) G^2}{(1-\beta_1)^2 \left(\beta_2^{t-2} + \epsilon^2\right)} + \frac{\alpha_t^2 G^2}{\beta_2^{t-1} + \epsilon^2} + \frac{\alpha_t (\alpha_{t-1} - \alpha_t) \beta_1 \sqrt{1 - \beta_1^{t-1}} G^2}{(1 - \beta_1) \sqrt{\beta_2^{t-1} + \epsilon^2} \sqrt{\beta_2^{t-2} + \epsilon^2}}. \quad (176)$$

$\square$

# H    DETAILS OF EXPERIMENTAL SETUPS

## H.1    CODE

Our implementation for the experiment is available at `https://anonymous.4open.science/r/adopt-iclr2024-submission-C7BE`.

## H.2    TOTAL AMOUNT OF COMPUTE

We run our experiments mainly on cloud GPU instances with $8\times$ A100. It took approximately 320 hours for our experiments in total.

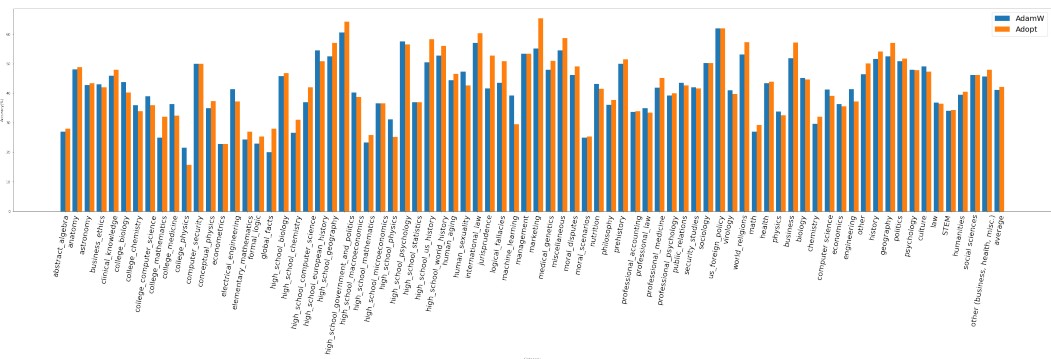

Figure 5: Comparison of MMLU scores for LLaMA-7B finetuned via instruction following using AdamW and ADOPT.

### H.3 LICENSE OF ASSETS

**Datasets:** The MNIST database is downloaded from `http://yann.lecun.com/exdb/mnist`, which is license-free. The terms of access for the ImageNet database is provided at `https://www.image-net.org/download`. The dataset of Stanford Alpaca is CC BY NC 4.0 (allowing only non-commercial use).

**Pretrained models:** The pretrained model of LLaMA is provided under GNU General Public License v3.0.

**Simulator:** MuJoCo is provided under Apache License 2.0.

**Code:** Our implementation of ImageNet classification is based on the Torchvision's official training recipe provided at `https://github.com/UiPath/torchvision/tree/master/references/classification`. Torchvision is provided under BSD 3-Clause License. We use the official implementation of NVAE provided at `https://github.com/NVlabs/NVAE`, whose license is described at `https://github.com/NVlabs/NVAE/blob/master/LICENSE`.