# OpenReview forum: "ADOPT: Modified Adam Can Converge with the Optimal Rate with Any Hyperparameters"
_ICLR.cc/2024/Conference — Submitted to ICLR 2024_

### Official Review · Reviewer_vPft · 2023-10-30

**Soundness:** 2 fair
**Presentation:** 3 good
**Contribution:** 2 fair
**Rating:** 5
**Confidence:** 4

**Summary:**

The authors provide a new algorithm as a variant of Adam. Different from Adam that needs to carefully choose the hyperparameters to ensure convergence, ADOPT can converge at the optimal rate with any hyperparameters.  Moreover, the authors relax the condition that gradients are uniformly bounded into the condition that the second-order moment of gradients is bounded. The experiments show that the proposed algorithm is compatible with the other algorithms.

**Strengths:**

1. The proposed algorithm can converge with arbitrary hyperparameters without divergence issues.

2. The proposed algorithm performs compatible with popular algorithms.

**Weaknesses:**

1.  For relaxing the condition of gradient assumption, it is misleading to claim that  "the convergence is established without the bounded stochastic gradient assumption". In fact, in the paper, the authors replace the uniformly bounded gradients with bounded second-order moments, which still bounds the gradients of the expected function. Further, [1] shows the convergence of Adam without the assumption of gradients uniformly bounded or second-order moments bounded.

2. In practice, people will use random shuffle instead of random sampling. However, the convergence results only hold for random sampling. In fact, there is a counter-example for the proposed algorithm that can not converge when we use a random shuffle. Let $\beta1 = \beta2 = 0$, and $f_1(x) = 1.9x$ and $f_2(x) = f_3(x) = -x$. With the constraint that $x \in [-1,1]$ the optimal solution should be 1 instead of -1. Thus, although the algorithm can converge in the random sampling setting, it does not deal the case in the practical case.


[1] Li, Haochuan, Ali Jadbabaie, and Alexander Rakhlin. "Convergence of Adam Under Relaxed Assumptions." arXiv preprint arXiv:2304.13972 (2023).

**Questions:**

See weakness.

---

> ### Author Response · Authors · 2023-11-20
> **Response to Reviewer vPft**
>
> Thank you for your insightful comments.
> We will answer your questions to address your concerns.
> Please also refer to the general response, in which the revision during the rebuttal period is explained.
>
> **Our assumptions seem stronger than existing analyses on Adam**
>
> As you mentioned, our Assumption 4 requires the true gradient $\nabla f$ to be bounded although the stochastic gradient $g$ does not need to be bounded.
> Hence, the difference between Assumptions 4 and 5 is that Assumptions 4 does not assume the gradient noise $\|| g - \nabla f \||$ to be bounded; instead, it assumes the variance of the stochastic gradient, i.e., $\mathbb{E} [ \|| g - \nabla f \||^2 ]$, to be bounded, which is weaker than the bounded noise assumption.
>
> Although Li et al. (2023) do not assume the true gradient $\nabla f$ to be bounded, they instead assume that the gradient noise $\|| g - \nabla f \||$ is bounded, which is not assumed in our analysis.
> They also provide a result where the bounded noise assumption is relaxed to a sub-Gaussian noise norm assumption, but the convergence rate gets a little worse in that case.
>
> Therefore, our assumptions are not quite stronger in total compared to existing analyses on Adam (e.g., Li et al. (2023)), and our result of ADOPT's optimal convergence without depending hyperparameters is much stronger than existing results on Adam, which require problem-dependent tuning of hyperparameters.
> We also discuss the relation to existing analyses in detail in the general response and Appendix A in the revised paper, so please also refer to them.
>
> **Counter example in the case of random shuffling**
>
> Thank you for pointing it out.
> We did not recognize that there is a counter example where ADOPT cannot converge when applying without-replacement sampling (a.k.a. random shuffling) instead of with-replacement sampling.
> Since random shuffling violates Assumption 2, it is out of scope of our theory; hence the existence of such a counter example does not contradict to our theoretical findings.
> However, as you mentioned, practitioners often use random shuffling, so we have come to think that the existence of a counter example in the random shuffling setting should be mentioned in the paper.
> Therefore, we have added a discussion on it in Appendix B in the revised version.
>
> We would like to emphasize again that the non-convergent issue in such a counter example can be easily avoided by using with-replacement sampling.
> Moreover, the difference between with- and without-replacement sampling becomes negligible when the dataset size is large enough, so it does not affect the practical performance very much.
> In fact, our experiments except for the toy example are performed using without-replacement sampling, but divergent behaviors are not observed.
>
> We would be glad to respond to any further questions and comments that you may have.
>
> Thanks.

---

> ### Author Response · Authors · 2023-11-22
> **A Gentle Remainder to Reviewer vPft**
>
> Thank you again for your efforts in reviewing our paper and your constructive comments. The discussion period will end soon, so please let us know if you have further comments about our reply to your feedback.
>
> Thanks.

---

### Official Review · Reviewer_BAKG · 2023-10-31

**Soundness:** 3 good
**Presentation:** 3 good
**Contribution:** 3 good
**Rating:** 5
**Confidence:** 3

**Summary:**

This paper proposes a new Adam-variant algorithm that aims to fix the non-convergence issues of Adam. Specifically, it uses the second moment estimation at previous iteration to perform the preconditioning. The preconditioning is done on the gradient instead of on the momentum as in the case of Adam. The preconditioned gradient is used in the update of momentum, and the momentum (not preconditioned momentum) is used in the update of parameters. The authors show that the modified algorithm (namely ADOPT) can converge for all hyperparameters.

**Strengths:**

The paper is very well written and can be easily understood. It clearly explains the technical challenges arise in analyzing Adam, and how to resolve them. The resulting algorithm seems to be simple and intuitive following the analysis.

**Weaknesses:**

- The theoretical contribution is not very significant. Even though the bounded gradient assumption is relaxed, the paper requires a bound on the gradient norm squared. For example, the bounded gradient assumption is also not required in the recent work by Yushun Zhang et al.[1]. Besides this, the proof technique seems to be quite similar to that of Zhou et al. [2].

- The experimental gain seems to be marginal. For the results in Table 1 and Table 2, it is better to add standard deviations to clearly contrast the results of different algorithms.

[1] Adam Can Converge Without Any Modification On UpdateRules

[2] AdaShift: Decorrelation and Convergence of Adaptive Learning Rate Methods

**Questions:**

N/A

---

> ### Author Response · Authors · 2023-11-20
> **Response to Reviewer BAKG**
>
> Thank you for your insightful comments.
> We will answer your questions to address your concerns.
> Please also refer to the general response, in which the revision during the rebuttal period is explained.
>
> **On the assumptions of our analysis compared to existing ones**
>
> As you mentioned, our Assumption 4 is stronger than the growth condition adopted in Zhang et al. (2022).
> Zhang et al. (2022) instead assume that the stochastic gradient $g$ is Lipschitz continuous, which is a stronger assumption than ours, in which only the true gradient $\nabla f$ is assumed to be Lipschitz.
> Moreover, Zhang et al. (2022) focus on the finite-sum problems, whereas we deal with general nonconvex problems.
> Therefore, our assumptions are not quite stronger in total compared to existing analyses on Adam, and our result of ADOPT's optimal convergence without depending hyperparameters is much stronger than existing results on Adam, which require problem-dependent tuning of hyperparameters.
> We also discuss the relation to existing analyses in detail in the general response and Appendix A in the revised paper, so please also refer to them.
>
> **Proof technique seems to be similar to that of Zhou et al. (2019)**
>
> Zhou et al. (2019) only consider a single convex optimization problem in their analysis, but our analysis deals with general smooth nonconvex optimization problems; hence the scope of our analysis is much broader than theirs.
> Moreover, techniques used for the proofs are also very different, so we think that our theoretical contribution is not minor compared to Zhou et al. (2019).
> About the theoretical contribution, we have added a detailed discussion in the general response and Appendix A in the revised paper, so please also refer to it.
>
> **Experimental gain seems marginal and error bars are missing in Tables 1 and 2**
>
> In the revised version, we have added error bars in Tables 1 and 2, which may help to show that the experimental gain is not so marginal.
> In addition, to clearly show the superiority of ADOPT over Adam (and AMSGrad), we have revised the toy experiment.
> Please see the general response for a detailed experimental settings.
> This experiment clearly shows that Adam tends to fail to converge when the gradient noise is large, even if $\beta_2$ is chosen to be close to 1 (e.g., 0.999), whereas ADOPT can always converge to the solution without careful tuning of hyperparameters.
>
> We would be glad to respond to any further questions and comments that you may have.
>
> Thanks.

---

> ### Author Response · Authors · 2023-11-22
> **A Gentle Reminder to Reviewer BAKG**
>
> Thank you again for your efforts in reviewing our paper and your constructive comments. The discussion period will end soon, so please let us know if you have further comments about our reply to your feedback.
>
> Thanks.

---

### Official Review · Reviewer_nLrX · 2023-10-31

**Soundness:** 2 fair
**Presentation:** 3 good
**Contribution:** 4 excellent
**Rating:** 6
**Confidence:** 3

**Summary:**

The paper proposes ADOPT. Compared with Adam, ADOPT uses a decoupled second-order moment estimator and applies exponential averaging after calculating the adaptive update directions. A wide range of experiments show great potential for the proposed algorithm.

**Strengths:**

The proposed ADOPT is a novel algorithm that deserves the attention of deep learning researchers.  The experiments contain a wide range of machine-learning tasks and show promising results.

**Weaknesses:**

The major issue is in the theoretical analysis. I feel that both (5) and (14) are vacuous. From assumption 4, we know $||\nabla f||^2\le G^2$. However, (14) basically says
$$
\min_{t=1,\cdots,T}||\nabla f(\theta_t)||^2\le O\left(G^2\frac{\alpha L}{\epsilon}\right)
$$
In practice, the learning rate $\alpha$ is not too small, $\epsilon$ has the order of 1e-8, the Lipschitz constant $L$ is very large, so $\frac{\alpha L}{\epsilon}$ is often far greater than $1$. So, the upper bound in (14) is even larger than $G^2$, which is correct by assumption. Therefore, the theoretical analysis is not meaningful.

The authors may try to look at the theoretical analysis in Shi et al., 2020; Zhang et al., 2022 to derive more meaningful bounds.

**Questions:**

As ADOPT contains two changes compared to Adam, the authors should consider adding some ablation studies in section 5 to see which change has the most significant impact on the performance. Such studies are very informative for the future improvement of adaptive stepsize algorithms.

Also, in Figure 1 and 3, ADOPT has larger variances. Does this mean ADOPT is less robust than Adam?

Did authors consider adding the bias correction step in the algorithm?

---

> ### Author Response · Authors · 2023-11-20
> **Response to Reviewer nLrX**
>
> Thank you for your insightful comments. We will answer your questions to address your concerns.
> Please also refer to the general response, in which the revision during the rebuttal period is explained.
>
> **Convergence bounds are too loose**
>
> We agree that the bounds in the original submission were too loose to be meaningful, so we have revised the theorems to derive tighter bounds.
> We believe that this will address your main concern.
>
> **Ablation study on each algorithmic modification**
>
> Thank you for an insightful suggestion.
> We have added an ablation study on algorithmic modifications from Adam to ADOPT in Section 5.
> As can be seen in Figure 2 in the revised paper, ADOPT fails to converge if either of the two modifications (i.e., decorrelation between $g_t$ and $v$, and change of order of momentum and scaling operation) is not applied.
> This result aligns with the theoretical finding, and helps it to be more convincing.
>
> **Large variance of empirical results**
>
> In Figure 1 of the original submission, ADOPT seems to have high variance when $\sigma = 10$ compared to Adam and AMSGrad, but we do not think that it is not because ADOPT is less robust.
> First of all, Adam does not even converge to the correct solution, so the variance comparison is less meaningful.
> As for AMSGrad, its variance seems lower than ADOPT, but this is because $v_t$ of AMSGrad is non-decreasing in terms of $t$, so the effective learning rate tends to be lower than ADOPT.
> If the learning rate of ADOPT is set to a lower value, the variance of ADOPT will also decrease.
>
> In addition, ADOPT's high variance in Figure 3 of the original submission is just due to the nature of deep reinforcement learning (RL).
> In deep RL, the variance tends to be larger as the agent obtains high reward, so this phenomenon is quite natural.
>
> Moreover, in the revision during the rebuttal period, we have added error bars in Tables 1 and 2, and we observe that the variance of ADOPT is not higher than baselines (e.g., AdamW).
>
> **On the bias correction technique**
>
> In fact, we have tried to add the bias correction technique of Adam to our ADOPT, and it empirically worked well.
> However, it did not bring performance improvement (also did not harm the performance) in practice, so we exclude it from our implementation for simplicity.
> Our hypothesis on why the bias correction is less meaningful for ADOPT is that $v_t$ is initialized to $1$ for ADOPT to prevent it from being too small at the first parameter update, so the effect of bias correction is limited compared to Adam.
> Moreover, as for the momentum $m_t$, initialization with $0$ works like a warm-up of the learning rate, so it may have a positive effect for stable training especially in the early phase of optimization.
> Although there might be a better way to introduce a technique like the bias correction for ADOPT, we leave it for future work.
> We have added a discussion on the bias correction in Appendix C in the revised version.
>
> We would be glad to respond to any further questions and comments that you may have.
>
> Thanks.

---

> ### Author Response · Authors · 2023-11-22
> **A Gentle Reminder to Reviewer nLrX**
>
> Thank you again for your efforts in reviewing our paper and your constructive comments. The discussion period will end soon, so please let us know if you have further comments about our reply to your feedback.
>
> Thanks.

---

### Official Review · Reviewer_Y8m7 · 2023-11-05

**Soundness:** 2 fair
**Presentation:** 3 good
**Contribution:** 2 fair
**Rating:** 5
**Confidence:** 4

**Summary:**

The paper proposes a new adaptive gradient method called ADOPT, which addresses the non-convergence issue of popular methods like Adam and RMSprop.  The method modifies the calculation of second moment estimates and the order of momentum calculation and scaling operations. Extensive numerical experiments demonstrate that ADOPT achieves competitive or superior results compared to existing methods across various tasks.

**Strengths:**

The paper introduces a new adaptive gradient method ADOPT that is as easy as the implementation of Adam, and enjoys easy convergence proofs.

The paper gives in-depth analysis for the convergence of ADOPT with toy examples, in comparison with the failure cases of Adam.

The paper conducts comprehensive numerical experiments on various tasks, demonstrating the competitive performance of ADOPT
 compared to the widely used Adam.

**Weaknesses:**

First, the convergence of Adam has been established without any modification, e.g., Defossez et al. 2022 “A simple convergence proof of Adam and AdaGrad”, Wang et al. 2022 "Provable Adaptivity in Adam" and Zhang et al. 2022 "Adam Can Converge Without Any Modification On Update Rules". The convergence of a modified version of Adam is not significant from theoretical sense unless the ADOPT can beat the performance of Adam in practice.

From the empirical results, the performance of ADOPT is not superior over Adam very much. People may be reluctant to use ADOPT in practice. As for the title "convergence with any hyperparameters", the paper does not verify the performance of ADOPT is not sensitive to hyper-parameters in practice.

**Questions:**

See the weakness

---

> ### Author Response · Authors · 2023-11-20
> **Response to Reviewer Y8m7**
>
> Thank you for your insightful comments. We will answer your questions to address your concerns.
> Please also check the general response, in which the revision during the rebuttal period is explained.
>
> **On theoretical contribution**
>
> As you mentioned, some previous works (e.g., Zhang et al. (2022)) demonstrate that the original Adam can converge without any modifications.
> However, they require to choose hyperparameters (e.g., $\beta_1$ and $\beta_2$) in a problem-dependent manner, as we stated in Sections 1 and 2.2.
> This constraint is problematic even in a practical sense, because we do not have access to the problem-specific parameters (e.g., $L$ and $G$ in our paper), so we need to heuristically tune the hyperparameters for each problem.
> Hence, to assure the convergence in general cases, algorithmic modifications are still essential as pointed out by Reddi et al. (2018).
> Although there have been some investigations on modifying Adam to assure convergence (e.g., AMSGrad), such methods require additional assumptions (e.g., bounded stochastic gradient) for convergence guarantees.
> Our theoretical contribution is on demystifying the fundamental cause of divergence of Adam, and proposing a modification without imposing a strong assumption like bounded stochastic gradient.
>
> To clarify these points, we have added a detailed discussion in Appendix A in the revised paper, so please also refer to it for more details.
>
> **Empirical investigations on hyperparameter sensitivity**
>
> We agree that our original experiments lacked the comparison of hyperparameter sensitivity between Adam and ADOPT, so we have revised our toy experiment in Section 5 (see the general response for detailed settings).
> In the experiment, we observe that Adam's convergence is sensitive to the choice of $\beta_2$ depending on the problem setting, whereas our ADOPT successfully converge without depending on the hyperparameter choice.
>
> We would be glad to respond to any further questions and comments that you may have.
>
> Thanks.

---

> ### Author Response · Authors · 2023-11-22
> **A Gentle Reminder to Reviewer Y8m7**
>
> Thank you again for your efforts in reviewing our paper and your constructive comments. The discussion period will end soon, so please let us know if you have further comments about our reply to your feedback.
>
> Thanks.

---

### Author Response · Authors · 2023-11-20
**General Response to the Reviewers**

We thank all the reviewers for carefully reading our paper and giving insightful comments.
Based on the reviewers' feedback, we have updated the paper to address their concerns.
The updated part is highlighted with colored text.

**(1) We have clarified the relation to existing analyses on Adam's convergence** (Reviewers Y8m7, BAKG, vPft).

Some reviewers seemed to have a concern about the theoretical contribution of this paper compared to existing analyses on Adam's convergence.
To clarify our theoretical contribution, we have added a more detailed explanations about how our analysis relates to existing works in Appendix A.

First of all, our main contribution is that we show Adam can converge to a stationary point without depending on the hyperparameter choice by slightly modifying the algorithm.
Although some previous works (e.g., Zhang et al. (2022)) insist that Adam can converge without any modifications, they require to choose hyperparameters in a problem-dependent manner; hence, to assure the convergence in general cases, algorithmic modifications are essential as pointed out by Reddi et al. (2018).
There have been some investigations on modifying Adam to assure convergence (e.g., AMSGrad), but such methods require additional assumptions (e.g., bounded stochastic gradient) for convergence guarantees.

Second, the assumptions we use in the analysis are not stronger compared to existing works.
For example, Zhang et al. (2022) uses a growth condition $\mathbb{E}\left[\||g\||^2\right] \leq G_0^2+G_1^2\||\nabla f\||^2$.
It is true that this assumption is weaker than our Assumption 4, but they instead assume that the stochastic gradient $g$ is Lipschitz, whereas we only assume that the true gradient $\nabla f$ is Lipschitz.
Moreover, their analysis is limited to finite-sum problems, while ours can be applied to general smooth nonconvex problems.
Li et al. (2023) also deal with general smooth nonconvex problems, but they require an assumption that the norm of gradient noise $\|| g - \nabla f \||$ is almost surely bounded, which is not assumed in our analysis.
De ́fossez et al. (2022) shares the same assumptions with ours, but their convergence rate $\mathcal{O} ( \log T / \sqrt{T} )$ is worse than our $\mathcal{O} ( 1 / \sqrt{T} )$ rate.

**(2) Convergence bound has been improved** (Reviewer nLrX)

Reviewer nLrX pointed out that our derived convergence bounds is too loose to be meaningful.
We agree that point, so we have derived better bounds for both Theorems 1, 2, (and 3 in the appendix).
We believe that this revision address his/her concern.

**(3) Toy experiment has been revised to clearly show the superiority of ADOPT** (Reviewers Y8m7 and BAKG)

Some reviewers appeared to doubt empirical superiority of ADOPT to Adam, so we have revised the experimental setting of the toy example in Section 5 to clearly demonstrate the superiority.
In the revised version, we use a stochastic objective $f_t \left( \theta \right) = k^2 \theta$ with probability $1/k$, otherwise $f_t \left( \theta \right) = -k \theta$.
In this case, the global objective is $f \left( \theta \right) = \theta$ regardless of the choice of $k$.
Hence, $k ( \geq 1)$ controls a noise level of the stochastic gradient.
Under this setting, we empirically show that, when $k$ is large, Adam fails to converge to the correct solution even if $\beta_2$ is set to a value very close to 1 (e.g., 0.999).
On the other hand, our ADOPT can converge without depending on the hyperparameter choice.
This result aligns with theoretical findings, and clearly shows the superiority of ADOPT to Adam.
For more details, see the revised paper.

**(4) We have added references to random shuffling in the appendix** (Reviewer vPft)

Reviewer vPft pointed out that there is a counter example where ADOPT fails to converge, if without-replacement sampling (a.k.a. random shuffling) is used instead of with-replacement sampling for finite-sum problems.
Since random shuffling violates Assumption 2, it is out of scope of our theory; hence the existence of such a counter example does not contradict to our theoretical findings.
However, as Reviewer vPft stated, random shuffling is often used in practice, so we have come to think that the existence of a counter example in the random shuffling setting should be mentioned in the paper.
Therefore, we have added a discussion on it in Appendix B in the revised version.

We would like to emphasize again that the non-convergent issue in such a counter example can be easily avoided by using with-replacement sampling.
Moreover, the difference between with- and without-replacement sampling becomes negligible when the dataset size is large enough, so it does not affect the practical performance very much.
In fact, our experiments except for the toy example are performed using without-replacement sampling, but divergent behaviors are not observed.

Thanks.

---

### Meta-Review · Area_Chair_Kmxt · 2023-12-06

**Metareview:**

The reviewers noted the ambitious goal of the paper to discuss convegence of Adam for large ranges of hyperparameters. However, there are outstanding concerns of whether what is proved by the authors support the strong claim made by the authors.

A reviewer raised the question of constants in the bounds and whether they are enough to claim convergence. While the authors did modify the paper, the bound has a dependency with respect to epsilon of 1/epsilon^2. In practice epsilon is kept negligible, previous bounds have a dependency of ln(1/epsilon). Taking epsilon as 10^-8 and we see that the bounds cannot give any meaningful indication of convergence.

**Justification For Why Not Higher Score:**

the bounds that are proved do not justify the convergence claims in realistic settings.

**Justification For Why Not Lower Score:**

N/A

---

### Decision · Program_Chairs · 2024-01-16

Reject